# Differential requirement for BRCA1-BARD1 E3 ubiquitin ligase activity in DNA damage repair and meiosis in the *Caenorhabditis elegans* germ line

Qianyan Li[1,2], Arshdeep Kaur[1], Kyoko Okada[1], Richard J. McKenney[1,2], JoAnne Engebrecht[1,2] *

1 Department of Molecular and Cellular Biology, University of California Davis, Davis, California, United States of America, 2 Biochemistry, Molecular, Cellular and Developmental Biology Graduate Group, University of California Davis, Davis, California, United States of America

* jengebrecht@ucdavis.edu

**Data Availability Statement:** All relevant data are within the manuscript and its Supporting Information files.

## Abstract

The tumor suppressor BRCA1-BARD1 complex regulates many cellular processes; of critical importance to its tumor suppressor function is its role in genome integrity. Although RING E3 ubiquitin ligase activity is the only known enzymatic activity of the complex, the *in vivo* requirement for BRCA1-BARD1 E3 ubiquitin ligase activity has been controversial. Here we probe the role of BRCA1-BARD1 E3 ubiquitin ligase activity *in vivo* using *C. elegans*. Genetic, cell biological, and biochemical analyses of mutants defective for E3 ligase activity suggest there is both E3 ligase-dependent and independent functions of the complex in the context of DNA damage repair and meiosis. We show that E3 ligase activity is important for nuclear accumulation of the complex and specifically to concentrate at meiotic recombination sites but not at DNA damage sites in proliferating germ cells. While BRCA1 alone is capable of monoubiquitylation, BARD1 is required with BRCA1 to promote polyubiquitylation. We find that the requirement for E3 ligase activity and BARD1 in DNA damage signaling and repair can be partially alleviated by driving the nuclear accumulation and self-association of BRCA1. Our data suggest that in addition to E3 ligase activity, BRCA1 may serve a structural role for DNA damage signaling and repair while BARD1 plays an accessory role to enhance BRCA1 function.

## Author summary

BRCA1-BARD1 is a E3 ubiquitin ligase, which modifies proteins by the addition of the small protein ubiquitin. While mutations that disrupt E3 ligase activity and stability of the BRCA1- BARD1 complex lead to a predisposition for breast and ovarian cancer, the specific requirement for E3 ligase activity in tumor suppression is not known. Here we probe the function of E3 ligase activity and BARD1 in the maintenance of genome integrity by engineering point mutations that disrupt E3 ligase activity in *C. elegans* BRCA1 as well as

**Funding:** Research in JE lab is funded by National Institutes of Health grants GM103860 and GM103860S1. Research in RJM lab is funded by National Institutes of Health grant 2R35GM124889-06. The funders had no role in study design, data collection and analysis, decision to publish, or preparation of the manuscript.

**Competing interests:** The authors have declared that no competing interests exist.

a null mutation in BARD1. While E3 ligase activity is important for genome integrity, the complex likely plays additional roles besides ubiquitylating proteins. Further, our data suggest that BRCA1 is the key functional unit of the complex while BARD1 is an accessory partner that enhances BRCA1's function. These findings may help explain why there is a higher prevalence of cancer-causing mutations in BRCA1 compared to BARD1.

## Introduction

BReast CAncer susceptibility gene 1 (BRCA1) and its obligate partner BARD1 (BRCA1 Associated RING Domain protein 1) are RING domain-containing proteins, which when mutated are linked to elevated incidence of breast and ovarian cancer [1–6]. The BRCA1-BARD1 complex functions in a myriad of cellular processes, including DNA damage repair, replication, checkpoint signaling, meiosis, chromatin dynamics, centrosome amplification, metabolism, and transcriptional and translational regulation [7–13]. BRCA1-BARD1 regulates these pathways presumably through ubiquitylation of substrates via its RING domains, which function as an E3 ubiquitin ligase. BRCA1 specifically interacts with E2-conjugating enzymes for ubiquitin transfer, while BARD1 greatly stimulates the E3 ligase activity of BRCA1 [14, 15].

BRCA1-BARD1 associates with multiple E2s to catalyze monoubiquitylation and several different polyubiquitylation chains on both itself and other protein substrates. Auto-ubiquitylation by BRCA1-BARD1 has been shown to occur via K6-linked poly-ubiquitin chains [16, 17]. BRCA1-BARD1 can also catalyze K48- and K63-linked polyubiquitylated chains when coupled with different E2s [18]. Even with the same E2s, BRCA1-BARD1 can generate K6-linked chains in auto-ubiquitylation reactions but monoubiquitylates protein substrates [16, 19, 20]. Currently, the significance and role of mono and polyubiquitylation mediated by BRCA1-BARD1 on itself or on substrates are unknown, nor the specific contributions of BRCA1 and BARD1 within the complex to these different ubiquitylation states.

Multiple potential BRCA1-BARD1 substrates have been identified; however, the physiological significance of most of these substrates is currently unknown [21]. One well established substrate in the context of DNA damage signaling and transcriptional regulation is histone H2A [20, 22–24]. Recent structural and molecular studies have led to mechanistic insights into recruitment of the complex to DNA damage sites and subsequent ubiquitylation of histone H2A. These studies have highlighted the targeting role of BARD1 to nucleosomes, where ubiquitylation of H2A by the complex promotes repair of DNA double strand breaks (DSBs). This most likely occurs by blocking recruitment of 53BP1, which promotes error-prone non-homologous end joining at the expense of homologous recombination [25–27]. However, the full spectrum of substrates and their relationship to regulation of different cellular processes are currently not known.

The role of BRCA1-BARD1 in DNA damage repair has been linked to its tumor suppressor function. Early studies suggested that BRCA1-BARD1 E3 ligase activity was not essential for either recombinational repair or tumor suppression. This conclusion was based on the analysis of a single isoleucine to alanine mutation at amino acid 26 (I26A) in the BRCA1 RING domain that abrogates its E3 ligase activity *in vitro* but maintains the stability of the BRCA1-BARD1 heterodimer, unlike many cancer-causing mutations that impair both E3 ligase activity and heterodimer stability [14, 18, 28–30]. Mice expressing the BRCA1[I26A] mutant protein were not prone to tumor formation and mutant cells were proficient for homology-directed repair of DSBs, suggesting that E3 ligase activity is not essential for tumor suppressor function [31, 32]. In depth biochemical analyses, however, have shown that the BRCA1[I26A] mutant still exhibits

residual E3 ligase activity when paired with a subset of E2 ubiquitin-conjugating enzymes in *in vitro* ubiquitin transfer assays. Mutation of two additional residues (leucine 63 and lysine 65 to alanines) within the BRCA1 RING domain in combination with I26A are required to completely abrogate E3 ligase activity *in vitro* without compromising the structural integrity of the complex [19]. These results suggest that BRCA1 harboring all three mutations is a true ligase dead mutant; however, the phenotypic consequence of this triple mutation has not been analyzed.

To define the requirement for E3 ligase activity *in vivo*, we focused on the *C. elegans* BRCA1 and BARD1 orthologs, BRC-1 and BRD-1. Previous analyses revealed that *C. elegans* BRC-1-BRD-1 is a functional E3 ubiquitin ligase that plays multiple roles in the cell [33]. Specifically, *brc-1* and *brd-1* mutants are sensitive to several DNA damaging agents, including irradiation, hydroxyurea, and crosslinkers, emphasizing a critical function in DNA damage repair [34–37]. Although BRC-1-BRD-1 is not essential for meiosis, mutation of *brc-1* or *brd-1* in combination with other mutations in meiotic genes have uncovered roles for the complex in meiotic inter-sister repair, repair pathway choice, crossover regulation, RAD-51 filament stability, and chromosome structure [38–44]. BRC-1-BRD-1 has also been shown to play important roles in the regulation of heterochromatin [45] and transcription [23]. Additionally, a recent study reported a role for BRC-1-BRD-1 E3 ligase activity in post-mitotic axon regeneration [46]. Thus, like the human complex, *C. elegans* BRC-1-BRD-1 plays pleiotropic roles *in vivo*, presumably through ubiquitylating different substrates. Here we analyzed the requirement for BRC-1-BRD-1 E3 ubiquitin ligase activity by generating worms expressing BRC-1 mutant proteins containing the corresponding single (I23A) and triple (I23A, I59A, R61A) mutations based on modeling with human BRCA1. Our data suggest that there are both E3 ligase-dependent and independent functions of BRC-1-BRD-1 in the context of DNA damage repair and meiosis. Intriguingly, E3 ligase activity and BRD-1 function can be partially bypassed by independently driving nuclear accumulation and self-association of BRC-1. Our data suggest that in addition to E3 ligase activity, BRC-1 may serve a structural role in DNA damage signaling and repair while BRD-1 plays an accessory role to enhance BRC-1 function.

## Results

### *brc-1(triA)* exhibits a more severe phenotype than *brc-1(I23A)*

The *C. elegans* orthologs of BRCA1 and BARD1 are structurally conserved with the same key domains as the human proteins: both BRC-1 and BRD-1 contain an N-terminal RING domain and C-terminal BRCT repeat domains [47]. The RING domains specify E3 ubiquitin ligase activity, while BRCT domains are phospho-protein interaction modules. Sequence alignment between human BRCA1 and *C. elegans* BRC-1 RING domains reveals that residues essential for E3 ligase activity in human BRCA1 (isoleucine 26, leucine 63, and lysine 65) correspond to amino acids isoleucine 23, isoleucine 59 and arginine 61 in *C. elegans* BRC-1 (Fig 1A). While not identical, these amino acids have similar chemical properties in terms of hydrophobicity and charge. To confirm that these BRC-1 residues structurally align with the human residues critical for ubiquitin transfer, we used AlphaFold to predict the structure of *C. elegans* BRC-1 RING domain (green), which was superimposed onto the NMR structure of the human BRCA1 RING domain (purple) using ChimeraX (Fig 1B) [14, 48, 49]. The predicted structure overlay is consistent with the sequence alignment in that isoleucine 23, isoleucine 59 and arginine 61 in BRC-1 are the structurally relevant counterparts of isoleucine 26, leucine 63 and lysine 65 in human BRCA1.

To probe the *in vivo* function of BRC-1 E3 ligase activity, we generated *C. elegans* mutants *brc-1(I23A)* [isoleucine 23 mutated to alanine] and *brc-1(triA)* [isoleucine 23, isoleucine 59,

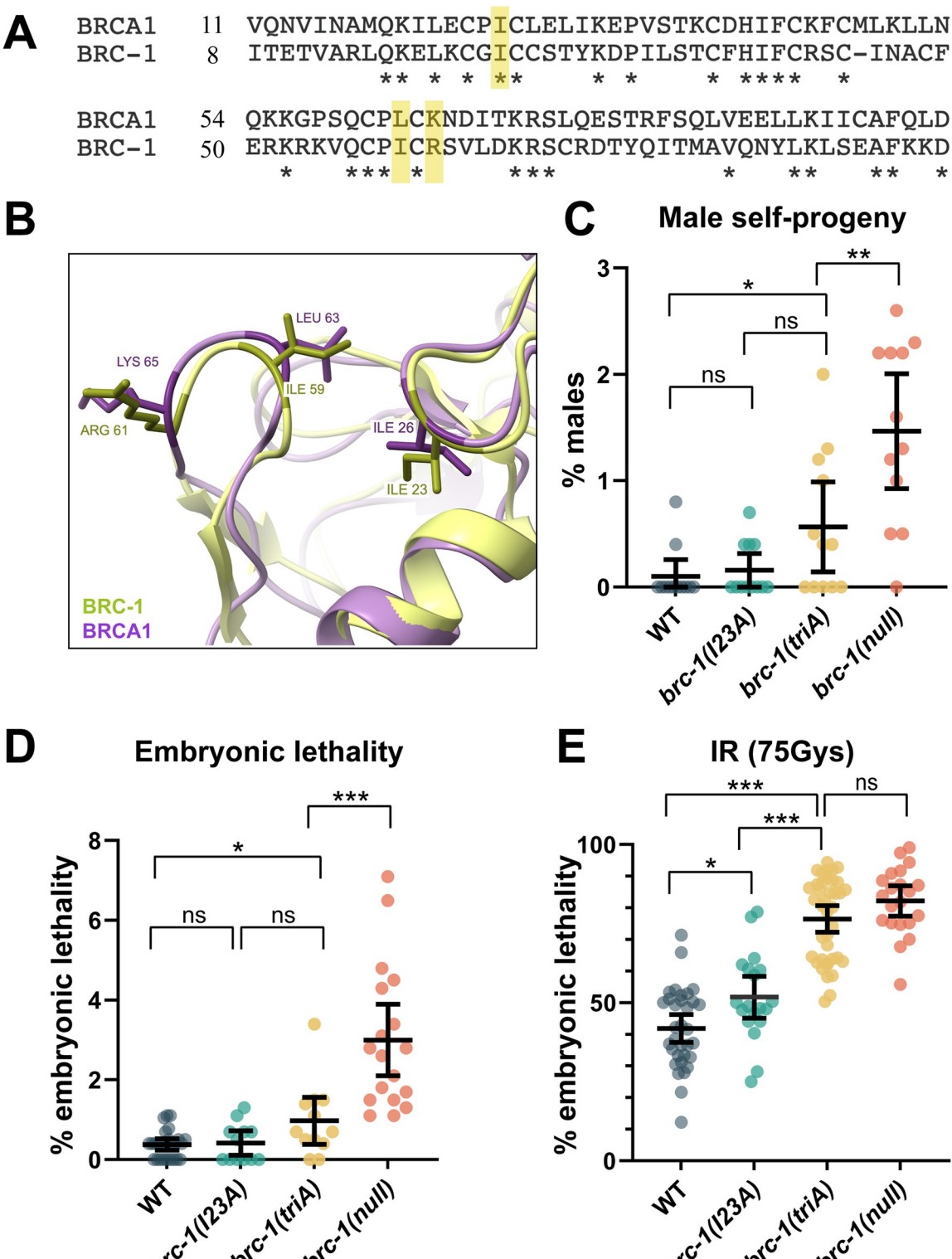

**Fig 1. Mutation of three key amino acids in the BRC-1 RING domain leads to a more severe phenotype than the single I23A mutation.** (A) Sequence alignment of RING domains from human and *C. elegans* BRCA1 orthologs reveals that amino acids isoleucine 23, isoleucine 59 and arginine 61 in *C. elegans* BRC-1 correspond to isoleucine 26, leucine 63 and lysine 65 in human BRCA1 (highlighted in yellow). (B) Predicted structure of BRC-1 RING domain (green) determined by AlphaFold, superimposed onto the NMR structure of the human BRCA1 RING domain (purple) showing the three amino acids occupy the same physical position. (C)

Male self-progeny, (D) embryonic lethality, and (E) embryonic lethality in the presence of 75Gys IR were examined in wild type, *brc-1 (I23A)*, *brc-1(triA)*, and *brc-1(null)* animals. Number of animals examined in (C): n = 12 for all genotypes; (D): WT n = 26; *brc-1(I23A)* n = 12; *brc-1(triA)* n = 12; *brc-1(null)* n = 18; (E) WT n = 38; *brc-1(I23A)* n = 19; *brc-1(triA)* n = 39; *brc-1(null)* n = 21. *** $p < 0.001$; ** $p < 0.01$; * $p < 0.05$; ns = not significant by Mann-Whitney.

arginine 61 mutated to alanines] at the endogenous *brc-1* locus using CRISPR-Cas9 genome editing and analyzed the mutant phenotypes with respect to meiosis and DNA damage repair. *brc-1* and *brd-1* mutants produce slightly elevated levels of male self-progeny (X0), a readout of meiotic X chromosome nondisjunction, have low levels of embryonic lethality under standard growth conditions, but display high levels of embryonic lethality after exposure to γ-irradiation (IR),which induces DNA DSBs [40, 42, 47]. For both male self-progeny and embryonic lethality under standard growth conditions, the *brc-1(I23A)* mutant produced similar levels to wild type, while *brc-1(triA)* worms gave rise to elevated levels compared to wild type but not to the extent of the *brc-1(null)* mutant [42] (Fig 1C and 1D). Following exposure to 75Gys of IR, *brc-1(I23A)* displayed higher levels of embryonic lethality compared to wild type, while *brc-1 (triA)* produced inviable progeny at levels comparable to those observed in the *brc-1(null)* mutant, suggesting that these residues are important when exogenous DNA damage is present (Fig 1E). Given the enhanced phenotype of *brc-1(triA)* relative to *brc-1(I23A)*, we also examined the consequence of mutation of isoleucine 59 and arginine 61 to alanines [*brc-1(I59A, R61A)*]. The double mutant behaved similarly to wild type (S1A and S1B Fig), suggesting that these residues do not affect function in an otherwise wild-type protein but are important in the context of the I23A mutation.

While BRC-1-BRD-1 plays only a subtle role in an otherwise wild-type meiosis as evidenced by the low levels of male self-progeny and embryonic lethality (Fig 1C and 1D), we previously showed that the BRC-1-BRD-1 complex plays a critical role when chromosome synapsis and crossover formation are perturbed by mutation of either pairing center proteins, which are required for pairing and synapsis of homologous chromosomes, or components of the synaptonemal complex (SC), the meiosis-specific protein structure that stabilizes homologous chromosome associations [41, 42]. To examine the phenotypic consequence of the I23A and triA RING domain mutants when meiosis is perturbed, we monitored embryonic lethality in the different *brc-1* mutants in combination with mutation of ZIM-1, a zinc finger pairing center protein that mediates the pairing and synapsis of chromosomes *II* and *III* [50]. *zim-1* mutants produce 60–70% inviable progeny due to random segregation of chromosomes *II* and *III* in meiosis resulting in the formation of aneuploid gametes. We observed a progressive increase in embryonic lethality in *brc-1(I23A); zim-1*, *brc-1(triA); zim-1*, and *brc-1(null); zim-1* mutants, consistent with our previous observation that *brc-1(triA)* is more severely impaired for function than *brc-1(I23A)*. These results also suggest that neither *brc-1(I23A)* nor *brc-1(triA)* are null alleles (Fig 2A).

In addition to enhancing embryonic viability, BRC-1 and BRD-1 stabilize the RAD-51 filament at mid to late pachytene in the *zim-1* mutant [41, 42]. The RAD-51 recombinase assembles on resected single strand DNA at DSBs and is essential for homology search and strand invasion during homologous recombination [51–53]. In wild type, RAD-51 filaments, visualized as nuclear foci by immunostaining, appear in the transition zone (leptotene/zygotene; zone 1), peak in early (zone 2) to mid pachytene (zone 3) and are removed in late pachytene (zones 3–4) [53] (Fig 2B, top). In mutants where crossover formation is blocked by defects in pairing or synapsis (e.g., *zim-1*), RAD-51 foci are extended into late pachytene and disappear by diplotene [42, 53, 54] (Fig 2B–2F). Removal of BRC-1 in this context results in a "RAD-51 dark region" at mid to late pachytene due to a defect in RAD-51 filament stability [42] (Fig 2C

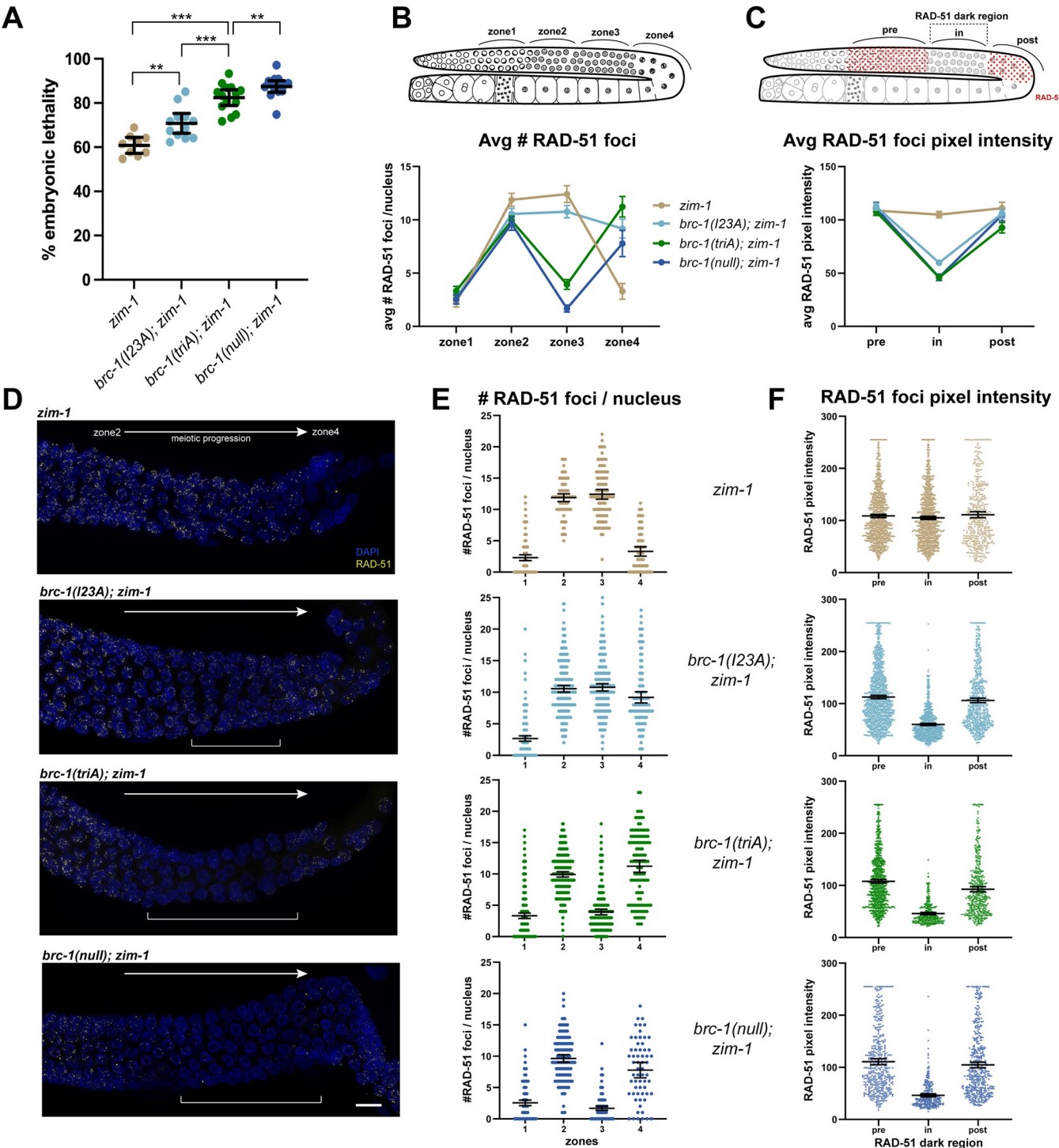

**Fig 2. *brc-1(I23A)* and *brc-1(triA)* mutants show differential defects in promoting progeny viability and RAD-51 filament stabilization in the *zim-1* mutant. (A)** Embryonic lethality of *brc-1* mutants in the *zim-1* mutant background. Emb of *brc-1(triA); zim-1* mutant (n = 14) is intermediate between *brc-1 (I23A); zim-1* (n = 12) and *brc-1(null); zim-1* mutants (n = 15), *zim-1* (n = 9). *** p < 0.001; ** p < 0.01 Mann-Whitney. **(B)** Cartoon of gonad indicating the zones analyzed for RAD-51 foci numbers across the meiotic region. In wild type, the zones correspond to 1 = transition zone; 2 = early pachytene; 3 = mid-late pachytene; 4 = late pachytene-diplotene. Graph depicts the average number of RAD-51 foci per nucleus quantified per zone from three germ lines of indicated genotypes. RAD-51 foci number only modestly declines in zone 3 in the *brc-1(I23A); zim-1* mutant. **(C)** Cartoon of gonad indicating regions analyzed for RAD-51 foci pixel intensity. Graph shows average pixel intensity of RAD-51 foci from pre, in and post RAD-51 dark region in three germ lines of the indicated genotypes. *brc-1(I23A); zim-1* contains nuclei with reduced RAD-51 foci intensity in the dark region but not to the extent of *brc-1(triA); zim-1* and *brc-1(null); zim-1* mutants. **(D)** Images showing part of the germ line from early/mid-pachytene (zone 2) to diplotene (zone 4) immunolabeled with RAD-51 antibody

(yellow) and counterstained with DAPI (blue). Brackets indicate the presence and location of RAD-51 dark region in the mutant germ lines, which is not as pronounced in the *brc-1(I23A); zim-1* mutant. Scale bar = 10μm (E) Scatter plot of number of RAD-51 foci per nucleus across the four zones. (F) Scatter plot of RAD-51 foci pixel intensity from pre, in and post RAD-51 dark regions in the germ lines. Mean and 95% CI are indicated for all data sets; statistical comparisons between genotypes are shown in S3 Table.

and 2D). This is manifested in a reduction in both RAD-51 foci numbers (zone 3; mid-late pachytene) and signal intensity ("in" dark region), followed by an increase in RAD-51 foci numbers (zone 4; late pachytene-diplotene) in the *brc-1(null); zim-1* mutant (Fig 2B–2F). To determine whether the RING domain mutants impair RAD-51 stabilization when meiosis is perturbed, we monitored RAD-51 foci number and signal intensity in the different *brc-1; zim-1* mutants (Fig 2B–2F). As with the increasing severity of embryonic lethality in the putative E3 ligase dead alleles in *zim-1* mutants, impairment of RAD-51 filament stability also showed a similar trend (Fig 2D). In zone 3, *brc-1(I23A); zim-1* had reduced numbers of RAD-51 foci compared to *zim-1*, but significantly higher numbers than observed in *brc-1(null); zim-1* (p = 0.0007 and p<0.0001, respectively). As meiosis progressed (zone 4), more RAD-51 foci were observed in *brc-1(I23A); zim-1* compared to *zim-1* alone (Fig 2B and 2E and S3 Table; p<0.0001). Additionally, average RAD-51 foci intensity in the *brc-1(I23A); zim-1* was significantly reduced in the RAD-51 dark region compared to *zim-1*, but not as reduced as in the *brc-1(null); zim-1* mutant ("in"; Fig 2C and 2F and S3 Table; p<0.0001). These findings suggest that *brc-1(I23A)* has a partial defect in RAD-51 filament stabilization leading to a smaller RAD-51 dark region in the germ line where RAD-51 foci numbers and intensity are reduced. In contrast to *brc-1(I23A); zim-1*, *brc-1(triA); zim-1* showed a severe reduction in average RAD-51 foci in zone 3, although not to the same extent as in *brc-1(null); zim-1* (p<0.0001). There was also a significant reduction in average RAD-51 foci intensity in the RAD-51 dark region in *brc-1(triA); zim-1* comparable to that observed in the *brc-1(null); zim-1* mutant ("in"; Fig 2C and 2D and 2F and S3 Table; p = 0.9999). Taken together, *brc-1(I23A)* has a weak phenotype, and *brc-1(triA)* is more severe, but still less severe compared to the *brc-1(null)*, suggesting that these mutations impair but do not completely abrogate the function of the complex when meiosis is perturbed.

## BRC-1$^{I23A}$ and BRC-1$^{triA}$ are impaired for E3 ubiquitin ligase activity *in vitro*

To determine whether BRC-1$^{I23A}$ and BRC-1$^{triA}$ are impaired for E3 ubiquitin ligase activity, we expressed and purified a chimeric form of the RING domains of BRC-1 and BRD-1 (BRD-1-BRC-1) in *E. coli*, modeled after studies of the human complex [18] (Figs 3A and S2A). The BRD-1-BRC-1 chimera was incubated in the presence of human E1 activating enzyme UBE1 (50% identical to *C. elegans* E1 UBA-1), E2 conjugating enzyme UbcH5c (94% identical to *C. elegans* E2 UBC-2), HA-ubiquitin (99% identical to *C. elegans* ubiquitin), and ATP and autoubiquitylation of the chimera was used as a readout for E3 ubiquitin ligase activity. The reaction was visualized by immunoblot using anti-HA antibodies and a characteristic ladder due to incorporation of multiple HA-ubiquitins into the chimera in the presence of ATP was observed, indicating robust auto-polyubiquitylation catalyzed by the BRD-1-BRC-1 RING (Fig 3B).

We next expressed and purified mutant chimeras harboring the I23A or triA (I23A, I59A, R61A) mutations (S2A Fig) and performed the auto-ubiquitylation assay. There was a significant reduction in the incorporation of HA-ubiquitin into the mutant complexes. While no polyubiquitylation was observed, there was reduced but detectable monoubiquitylation of both I23A and triA chimeras by an end point assay (I23A = 0.14±0.03, triA = 0.12±0.03 of

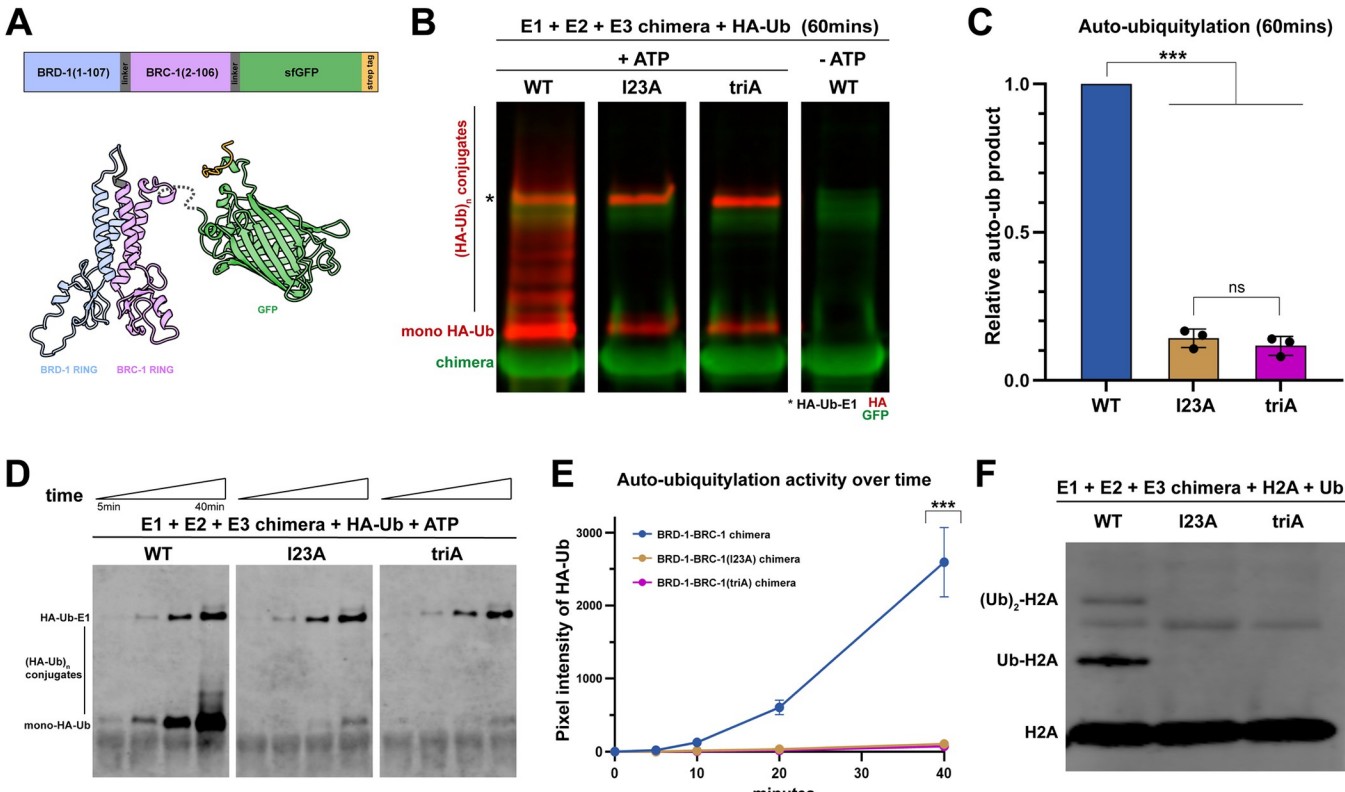

**Fig 3. BRD-1-BRC-1$^{I23A}$ and BRD-1-BRC-1$^{triA}$ chimeras are defective for E3 ubiquitin ligase activity *in vitro*.** (A) Construct and model based on AlphaFold of BRD-1-BRC-1 chimera: N-terminal BRD-1 RING domain (amino acids 1–107; blue), GGSGG linker (grey) and the BRC-1 RING domain (amino acids 2–106; purple) are connected to a superfold GFP (green) and strep II tag (orange) at the C terminus. Mutant chimera proteins contain either the single I23A or the I23A I59A R61A triple mutations (triA) in the BRC-1 RING, respectively. (B) Immunoblot showing auto-ubiquitylation (anti-HA-Ub, red) of BRD-1-BRC-1 chimera (anti-GFP, green) when incubated with E1, E2, HA-Ub and ATP for 60mins. *E1 incorporates HA-Ub (HA-Ub-E1) independently of E2 or E3s. Wild-type chimera promotes the formation of both auto-mono (mono HA-Ub) and polyubiquitylated (HA-Ub$_n$) conjugates while only reduced levels of auto-monoubiquitin BRD-1-BRC-1 were present using mutant chimeras. (C) Quantification of total HA-Ub signal at the end of 60mins showed that I23A and triA chimeras produced an average of 14% and 12% of total ubiquitylation, respectively, as compared to the wild-type chimera (Student T test, *** p<0.001). The difference between I23A and triA is not significant (ns) by Student T test, p = 0.55. (D) Time-course experiment to compare the kinetics of E3 ligase activity of the wild-type and mutant chimeras. Immunoblot showing HA-Ub signal at 5, 10, 20, and 40mins after the respective chimera was incubated with E1, E2, HA-Ub and ATP. (E) Quantification of HA-Ub signals plotted against time in wild-type and mutant chimeras (At 40mins: I23A = 0.041±0.013, triA = 0.028±0.016 of wild-type auto-ubiquitylation; Student T test, *** p<0.001). (F) Immunoblot of *C. elegans* ubiquitin incorporation into human histone H2A (anti-H2A) by WT and mutant chimeras; only WT was able to transfer ubiquitin to histone H2A protein to generate mono (Ub-H2A) and di-ubiquitin H2A (Ub)$_2$-H2A, but no ubiquitin incorporation into H2A was observed with either the I23A or I23A, I59A, R61A mutant chimeras.

wild-type auto-ubiquitylation; p<0.0001 comparing wild type and mutants using Student T test; Fig 3B and 3C). Time course analyses with decreased concentrations of reaction components revealed a significant reduction in the kinetics of ubiquitin incorporation of the I23A and triA chimeras (p<0.0001 comparing wild type and mutants using Student T test; Fig 3D and 3E). The physiological significance of the residual auto-monoubiquitylation observed in the mutant chimeras is not clear as it was also observed in reactions lacking the E2 conjugating enzyme (S2B Fig).

Histone H2A is a known physiological substrate of human BRCA1-BARD1 [22, 24] and *C. elegans* BRC-1-BRD-1 can ubiquitylate H2A *in vitro* [23]. We next determined whether the mutant chimeras could catalyze the incorporation of HA-ubiquitin into human histone H2A (90% identical to *C. elegans* H2A). Using antibodies directed against histone H2A or HA, we observed incorporation of mono and di-ubiquitin into H2A in the wild-type reaction; however, no ubiquitin incorporation into H2A was detected with either mutant chimera (Figs 3F

and S2C). From these experiments we conclude that BRC-1 harboring the I23A or the triA mutations are significantly impaired for E3 ubiquitin ligase activity under these *in vitro* conditions.

## Nuclear accumulation and BRC-1-BRD-1 interaction are differentially affected by BRC-1^I23A and BRC-1^triA

The finding that the *brc-1(I23A)* mutant had considerably weaker phenotypes in DNA damage repair and meiosis compared to the *brc-1(triA)* mutant but displayed similar impairment in E3 ubiquitin ligase activity *in vitro*, led us to examine the consequence of these mutations in more detail. We first monitored the localization of the mutant complexes using antibodies directed against BRD-1 [33]. BRC-1 and BRD-1 are mutually dependent for localization and are enriched in germ cell nuclei. In mitotic and early meiotic germ cells the complex is observed diffusely on chromatin and in foci. As meiosis progresses BRC-1-BRD-1 becomes associated with the SC and is then restricted to six small stretches on the six pairs of homologous chromosomes defined by the single crossover site [33, 40, 42] (Fig 4A). In the *brc-1(I23A)* mutant, BRD-1 displayed a similar localization pattern as wild type, although the intensity of the signal was weaker and was not as concentrated in the nucleus. Additionally, instead of six stretches, BRD-1 was concentrated into foci with some weak extensions in late pachytene/diplotene (Fig 4A and 4B). Nuclear accumulation of BRD-1 was further impaired in the *brc-1(triA)* mutant in proliferating germ cells through mid-pachytene where the protein was enriched in the cytoplasm (Fig 4A and 4B). At late pachytene and diakinesis in the *brc-1(triA)* mutant, BRD-1 was observed as weaker puncta in the nucleus, in addition to the cytoplasmic signal (Fig 4A). A similar localization pattern was observed by live cell imaging in the corresponding *brc-1* mutants expressing BRD-1::GFP at the endogenous locus [42] (S3A Fig). To determine whether BRC-1 is also cytoplasmic when E3 ligase activity is impaired, we examined the localization of BRC-1^triA in the C-terminally-tagged functional *brc-1::HA* allele [40]. As with BRD-1 in the *brc-1(triA)* background, BRC-1^triA::HA is also largely cytoplasmic and its function is impaired similarly to *brc-1(triA)* in response to IR (S3B Fig). These results suggest that nuclear accumulation of the complex is impaired in the E3 ligase defective mutants.

Given the reduced signal of BRD-1 observed by immunofluorescence in the *brc-1(I23A)* and *brc-1(triA)* mutants, we next examined steady state protein levels by immunoblot analysis. For these experiments we used worms expressing BRD-1::GFP, which also contain 3 copies of the FLAG epitope, and monitored protein levels using anti-FLAG antibodies. We observed a modest reduction of BRD-1 steady state levels in the *brc-1(I23A)* and *brc-1(triA)* mutants compared to wild type (*brc-1(I23A) brd-1::gfp* = 81.5±6.2% of *brd-1::gfp*, p = 0.0076; *brc-1(triA) brd-1::gfp* = 74.6±4.5% of *brd-1::gfp*, p = 0.0004) (Fig 4C and 4D). However, there was no significant difference between the two mutants (p = 0.19), suggesting that the difference in phenotypes observed *in vivo* is not a consequence of greatly altered steady state protein levels in the mutants but likely reflects the change in cellular distribution.

It has been reported that mutation of either I26 or I26, L63, K65 does not alter the interaction between human BRCA1 and BARD1 [19]. To determine whether this is also the case for the *C. elegans* orthologs, we examined interaction between full length BRC-1 and BRD-1 using the yeast two-hybrid system, which has previously been used to demonstrate interaction between these two proteins [47]. As expected, an interaction was detected between wild-type BRC-1 and BRD-1 using *his3* expression as a reporter by monitoring growth on medium lacking histidine. We observed slightly less growth with the BRC-1^triA mutant, suggesting that while BRC-1^triA interacts with BRD-1, there is some impairment (Fig 4E). Quantitative analysis of an independent reporter, β-galactosidase, revealed a ~50% decrease in the interaction

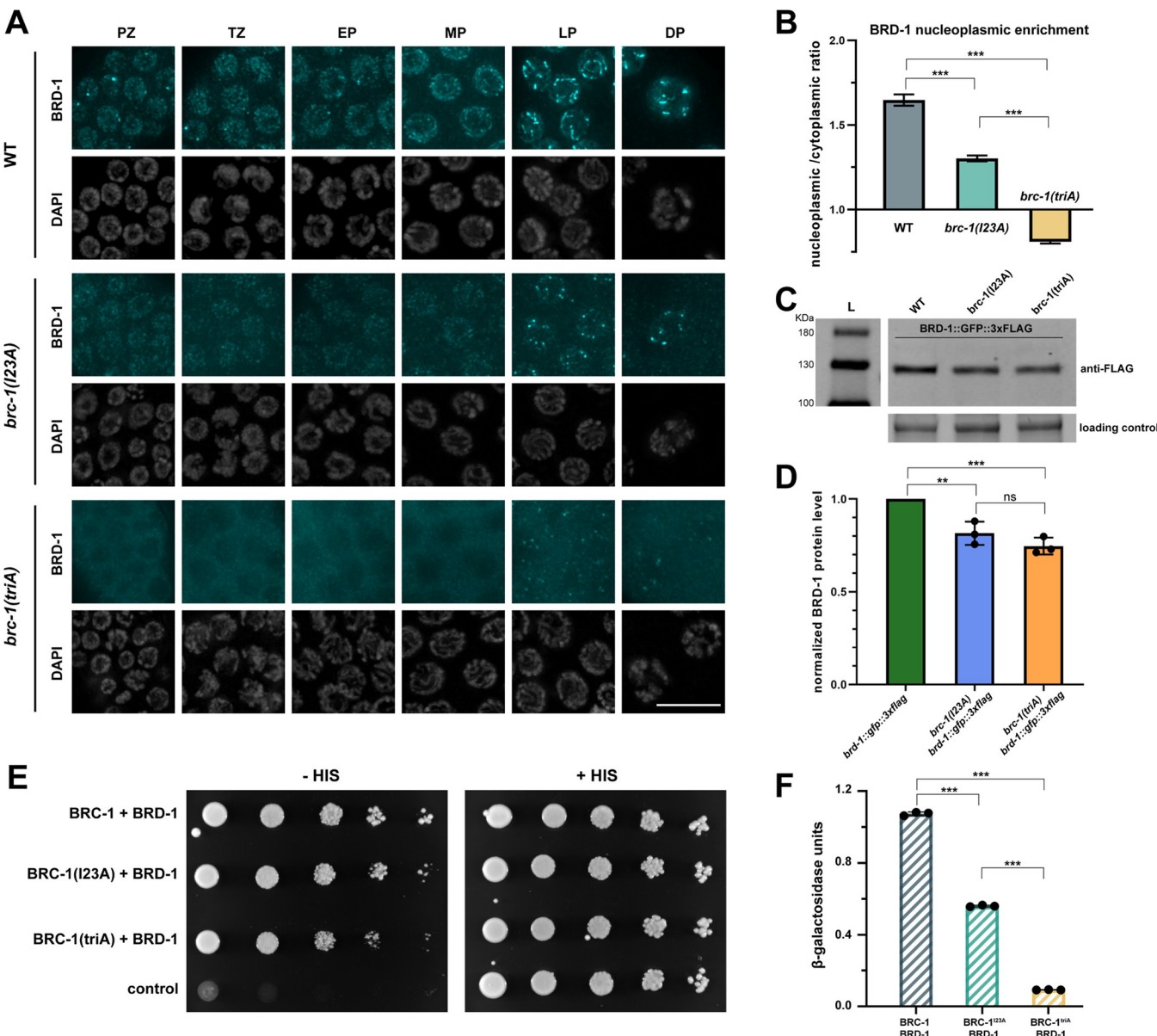

**Fig 4. Nuclear accumulation and BRC-1-BRD-1 interaction are differentially affected by BRC-1$^{I23A}$ and BRC-1$^{triA}$ mutations.** (A) Images of germline nuclei showing BRD-1 immunolabeling (cyan) by anti-BRD-1 antibodies and corresponding DAPI. PZ = proliferative zone; TZ = transition zone; EP = early pachytene; MP = mid pachytene; LP = late pachytene; DP = diplotene stages in the germ line. Scale bar = 10μm. (B) Graph shows nucleoplasmic to cytoplasmic ratio of BRD-1 signal. 40 nuclei from proliferative zone to mid pachytene from 3 germ lines were analyzed. Mann Whitney *** $p<0.001$. (C) Immunoblot of BRD-1::GFP::3xFLAG from whole worm extracts. BRD-1::GFP::3xFLAG migrates slower than its predicted size (112 kDa). (D) Quantification of BRD-1::GFP::3xFLAG steady state levels in the *brc-1* mutants normalized to wild type from 3 independent experiments. Student T test *** $p<0.001$. (E) Yeast 2-hybrid interaction between BRC-1 and BRD-1 as measured by growth on medium lacking histidine (-HIS) with +HIS as control. (F) Relative β-galactosidase activity assay showing reduced interaction between mutant BRC-1 (I23A and triA) and BRD-1 in corresponding yeast strains. Student T test *** $p<0.001$.

between BRC-1$^{I23A}$ and BRD-1 and a ~90% interaction defect between BRC-1$^{triA}$ and BRD-1 (Fig 4F). Thus, mutations in BRC-1 residues important for E3 ligase activity also affect interaction with BRD-1 in the yeast two-hybrid system. Whether these mutations impair interaction *in vivo* is unknown, but this potential interaction defect may contribute to the more severe phenotype of *brc-1(triA)*.

## GFP fused to E3 ligase defective BRC-1 restores nuclear localization and partially rescues defects in DNA damage repair

In the course of our experiments we discovered that fusion of GFP to the N-terminus of the E3 ligase impaired *brc-1* mutants had less severe phenotypes in response to DNA damage compared to the non-tagged alleles, although only *gfp::brc-1(triA)* vs. *brc-1(triA)* reached statistical significance and was investigated further (p<0.0001; Fig 5A). Rescue in viability was specific to N-terminally tagged BRC-1, near where BRC-1 and BRD-1 interact [14], as C-terminal fusion of GFP to BRD-1 did not rescue embryonic lethality following IR in the E3 ligase defective mutants (S3E Fig). Rescue by N-terminal GFP fusion was not a consequence of a change in BRC-1 expression, as there was no difference in steady state protein levels (S3F Fig). Interestingly, localization by live cell imaging revealed that unlike BRC-1^triA::HA or BRD-1 (visualized by BRD-1 antibodies or BRD-1::GFP live cell imaging) in the *brc-1(triA)* mutant, GFP:: BRC-1^triA was enriched in the nucleus and concentrated to six stretches at late pachytene/diplotene (-IR; Figs 5B, 5C and S3), consistent with nuclear localization being important for function. As BRC-1-BRD-1 becomes enriched in nuclear foci in response to DNA damage [33, 42] and we observed improvement of function when GFP was fused to BRC-1^triA following IR

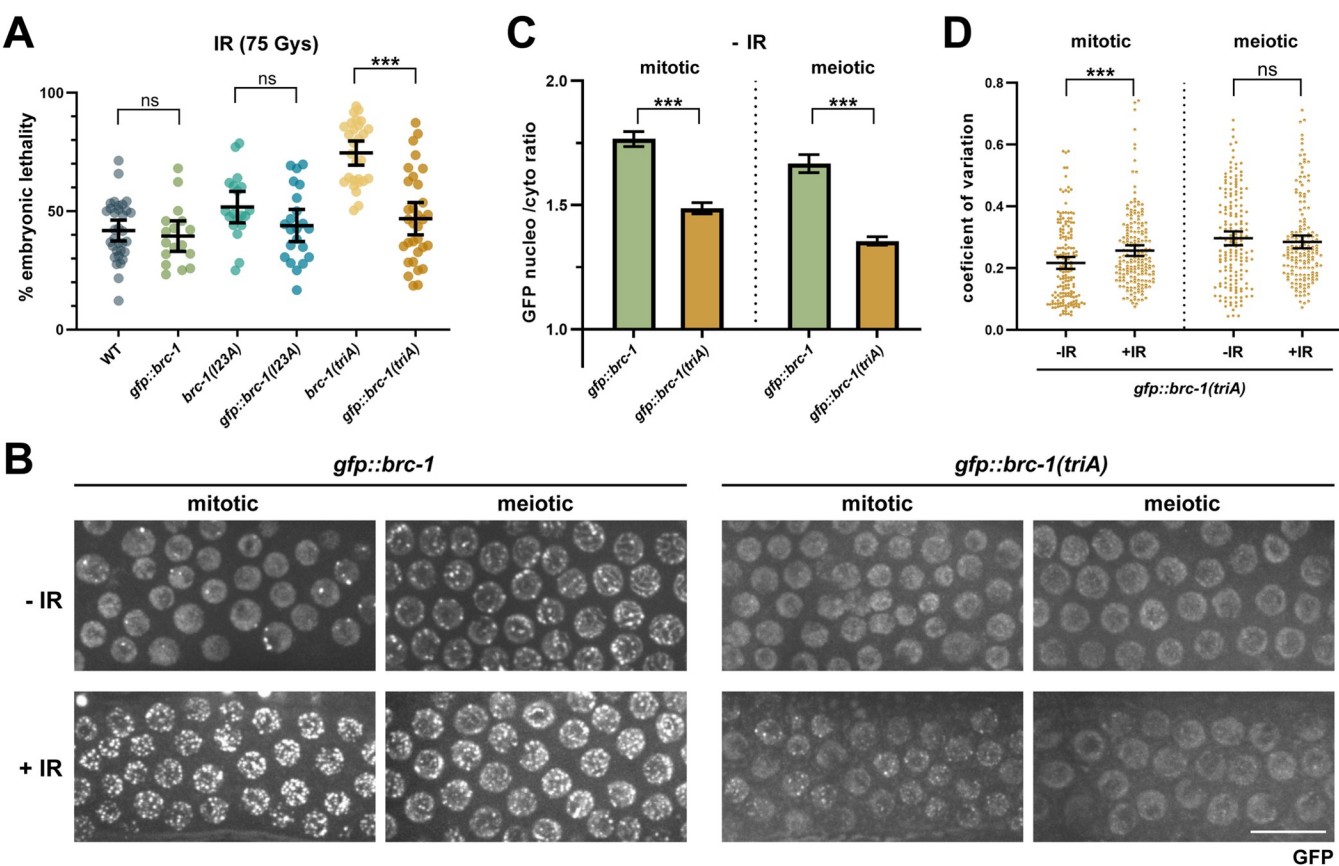

**Fig 5. Nuclear accumulation of BRC-1 with impaired E3 ligase activity promotes viability in response to DNA damage.** (A) Embryonic lethality in the presence of 75Gys IR was examined in wild type (n = 33), *gfp::brc-1* (n = 17), *brc-1(I23A)* (n = 19), *gfp::brc-1(I23A)* (n = 22), *brc-1(triA)* (n = 28), and *gfp::brc-1 (triA)* (n = 32) animals. *** p < 0.001 Mann-Whitney. (B) Images from live worms of mitotic (proliferative zone) and meiotic (early-mid pachytene) germ cells expressing GFP::BRC-1 or GFP::BRC-1^triA in the absence (-IR) and presence (+IR) of 75Gys radiation. Scale bar = 10μm. (C) Graph shows nucleoplasmic to cytoplasmic ratio of GFP signal in *gfp::brc-1* and *gfp::brc-1(triA)* strains. A minimum of 60 nuclei from 3 germ lines were analyzed. (D) Coefficient of variation (CV) for GFP::BRC-1^triA fluorescence to reflect changes in localization (foci formation) in response to IR in mitotic and meiotic nuclei in the germ line; five germ lines were analyzed for each genotype. Statistical comparisons between—and + IR *** p < 0.001 Mann-Whitney.

exposure, we examined localization of the mutant complex in response to IR-induced DNA damage. Both GFP::BRC-1 and GFP::BRC-1triA remained nucleoplasmic and concentrated into foci following exposure to IR, although foci intensity was considerably weaker in *gfp::brc-1(triA)* compared to *gfp::brc-1* (+IR; Fig 5B). We also observed weak BRD-1::GFP nuclear foci and very little nucleoplasmic enrichment in *brc-1(triA)* in response to IR in proliferating germ cells (S3C Fig). These results suggest that nucleoplasmic BRC-1-BRD-1 is important for function in response to DNA damage.

In contrast to mitotically-dividing germ cells, no GFP::BRC-1triA or BRD-1::GFP foci in the *brc-1(triA)* mutant were observed in meiotic nuclei. However, only GFP::BRC-1triA was nucleoplasmic, while no nuclear enrichment of BRD-1::GFP fluorescence was observed in *brc-1 (triA)* (Figs 5B and S3C). To quantitatively compare the extent of nuclear foci formation, we calculated the coefficient of variation (CV), which describes the dispersion of pixel intensity values from a region of interest around the mean pixel intensity such that nuclei with more foci above the nucleoplasmic GFP signal will have higher CV values, whereas nuclei with few foci will have lower CV values [55]. In mitotically-dividing *gfp::brc-1(triA)* mutant germ cells there was a significantly higher CV in IR treated worms compared to -IR worms; however, in meiotic cells there was no difference following IR treatment (Fig 5D).

## E3 ligase activity is essential for recruitment of BRC-1 to DSBs in meiotic cells independent of meiotic chromosome structure

Significantly fewer GFP::BRC-1triA foci were observed in meiotic nuclei compared to mitotic germ cell nuclei after exposure to IR, suggesting that meiosis is more sensitive to loss of E3 ligase activity in recruiting the complex to DNA damage sites (Fig 5B and 5D). To probe the requirement for BRC-1-BRD-1 E3 ligase activity in recruitment of the complex to meiotic DSBs when meiosis is perturbed, we monitored GFP::BRC-1triA localization in the *syp-1* mutant, where homologous chromosomes fail to synapse and no crossovers are formed [53]. As we previously reported, there were extensive GFP::BRC-1 nuclear foci in the *syp-1* mutant; these foci presumably represent meiotic recombination sites that are delayed in repair due to the absence of a homologous repair template in addition to activation of a meiotic checkpoint that upregulates meiotic DSB formation [42, 56–59]. In contrast to GFP::BRC-1, essentially no GFP::BRC-1triA foci were observed in the *syp-1* mutant (Fig 6A and 6F). This result is consistent with E3 ligase activity being important for accumulation of BRC-1-BRD-1 at sites of meiotic recombination.

To determine whether the more severe defect in recruiting GFP::BRC-1triA to sites of recombination in meiotic germ cells compared to the exogenous DNA damage induced sites in proliferating germ cells was a consequence of barriers imposed by the specialized meiotic chromosome structure, we examined the requirement for BRC-1-BRD-1 E3 ligase activity in recruitment of the complex to DSBs in mutants defective in the formation of the chromosome axes. To that end, we monitored the localization of GFP::BRC-1 and GFP::BRC-1triA when axis formation was impaired by mutation of the HORMA domain protein, HIM-3. HIM-3 is an axis component and is required for homolog pairing and synapsis and promotes crossover formation by biasing recombination to the homologous chromosome instead of the sister chromatid [60, 61]. While GFP::BRC-1 is recruited to both foci and the occasional track in the *him-3* mutant, as we have observed previously in mutants defective in meiotic recombination (e.g., *spo-11*, *mre-11*, *msh-5*) [41, 42], only tracks but no GFP::BRC-1triA foci were observed in the *him-3* mutant (Fig 6B and 6F). Quantification showed that the average number of GFP::BRC-1 foci was lower in the *him-3* mutant compared to *syp-1*, suggesting that repair is more efficient in *him-3* mutants due to release of the barrier to inter-sister repair (Fig 6F). We also examined

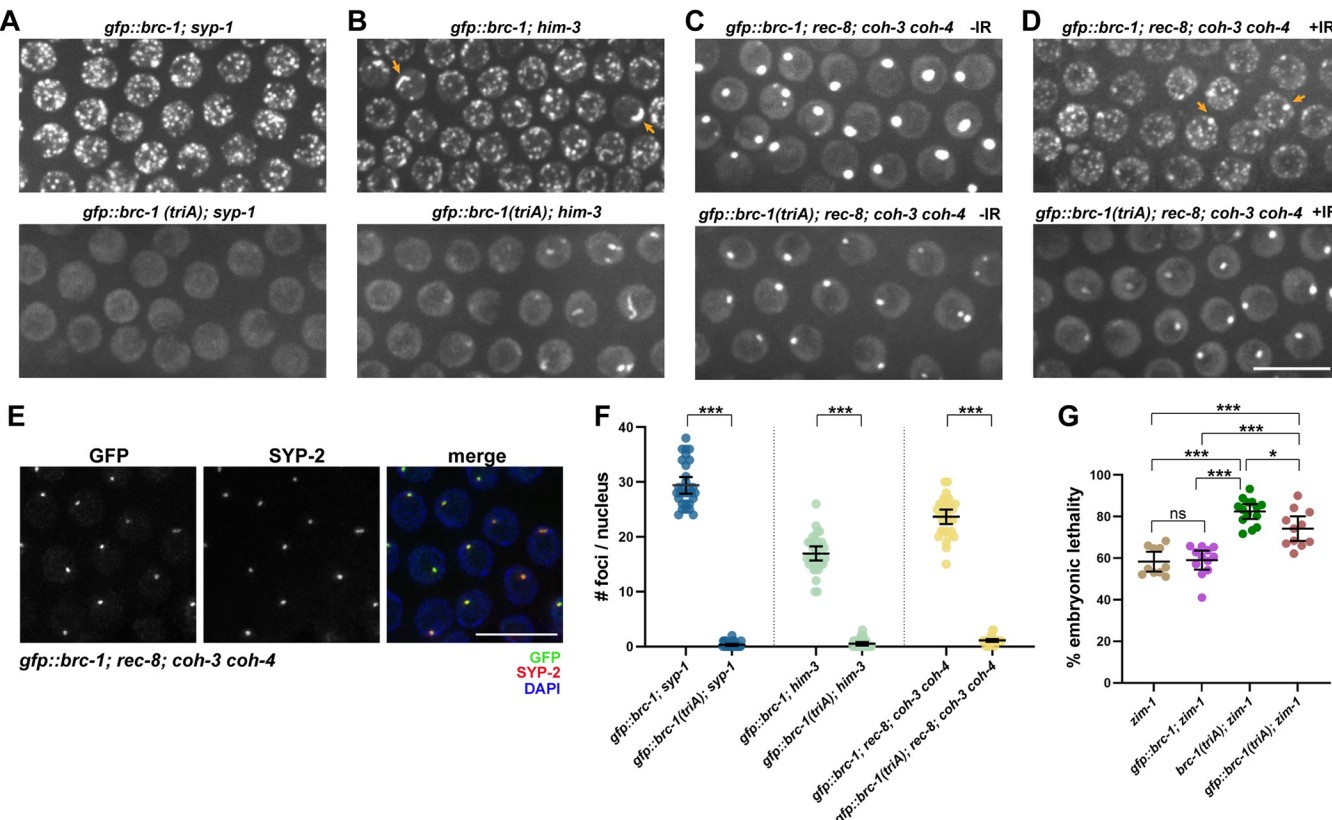

**Fig 6. BRC-1-BRD-1 E3 ligase activity is essential for recruitment of the complex to DSBs on meiotic chromosomes.** Images from live worms of meiotic (early-mid pachytene) germ cells expressing GFP::BRC-1 or GFP::BRC-1[triA] in *syp-1* (A), *him-3* (B), *rec-8; coh-3 coh-4* (C) without IR, and (D) *rec-8; coh-3 coh-4* mutants in the presence of 75Gys IR. Scale bar = 10μm. (E) Images of fixed mid-late pachytene nuclei labeled with anti-SYP-2 antibodies (red), BRC-1 (GFP fluorescence; green) and counterstained with DAPI (blue); scale bar = 10μm. (F) Number of GFP::BRC-1 or GFP::BRC-1[triA] foci observed in the different mutants; a minimum of 3 germ lines from half-projections were scored. Mann-Whitney *** p<0.001. (G) Embryonic lethality in *zim-1* (n = 10), *gfp::brc-1; zim-1* (n = 12), *brc-1(triA); zim-1* (n = 14), *gfp::brc-1(triA); zim-1* (n = 11) animals. Mann-Whitney * p < 0.05.

the consequence of impairing meiotic chromosome cohesion and hence axis formation by mutation of the meiosis-specific cohesin kleisin subunits, REC-8, COH-3 and COH-4 [62, 63]. Unlike *syp-1* and *him-3* mutants, *rec-8; coh-3 coh-4* triple mutants are not competent for meiotic DSB formation and therefore no GFP::BRC-1 or GFP::BRC-1[triA] foci were detected [63] (Fig 6C). However, we observed a bright aggregate of both GFP::BRC-1 and GFP::BRC-1[triA] in the *rec-8; coh-3 coh-4* mutant (Fig 6C). These aggregates co-labeled with antibodies against SYP-2, a central region component of the SC [53], suggesting that the aggregates are polycomplexes, SC-like structures formed independently of chromosomes (Fig 6E). We next monitored recruitment of GFP::BRC-1 and GFP::BRC-1[triA] to DNA breaks induced by IR in the *rec-8; coh-3 coh-4* mutant and while we observed abundant GFP::BRC-1 foci, no GFP::BRC-1[triA] foci were detected (Fig 6D and 6F). These results suggest that the underlying meiotic chromosome cohesion and axis do not impose a special requirement for BRC-1-BRD-1 E3 ligase-dependent recruitment of the complex to DSBs.

We next examined the phenotypic consequence of the inability to recruit nuclear GFP::BRC-1[triA] to meiotic foci by examining progeny viability in the *zim-1* mutant. We observed improved progeny viability of *gfp::brc-1(triA); zim-1* compared to *brc-1(triA); zim-1*, but not to the extent of what was observed in response to IR (Figs 6G and 5A). Thus, nuclear BRC-1-BRD-1 provides some function despite its inability to accumulate at DSBs. Taken together,

meiosis imposes a special requirement for BRC-1-BRD-1 E3 ligase activity in recruitment of the complex to DSBs.

## BRD-1 function can be partially bypassed by expressing GFP::BRC-1

The partial rescue of the E3 ligase impaired mutants by fusing GFP to BRC-1, but not to BRD-1, prompted us to explore the contribution of BRD-1 to the function of the complex. To that end, we constructed a putative null allele (*brd-1(null)*) by engineering multiple stop codons in the second exon of *brd-1* as described [64], as available alleles of *brd-1* are in-frame deletions C-terminal to the RING domain and helices where BRC-1 and BRD-1 interact [40, 42]. *brd-1 (null)* transcript was unstable, and no GFP fluorescence or BRD-1 protein by immunoblot was detected in *brd-1(null)* worms containing GFP and FLAG fused to the C-terminus of *brd-1 (null)*, providing evidence that it is a null allele (S4A–S4C Fig). Further, *brd-1(null)* was phenotypically indistinguishable from *brc-1(null)* for male self-progeny and embryonic lethality in the absence and presence of IR (S4D and S5A Figs).

We next examined the phenotype of *gfp::brc-1 brd-1(null)* and saw a weak rescue of *brd-1 (null)* embryonic lethality following exposure to IR (Fig 7A). Consistent with this, GFP::BRC-1

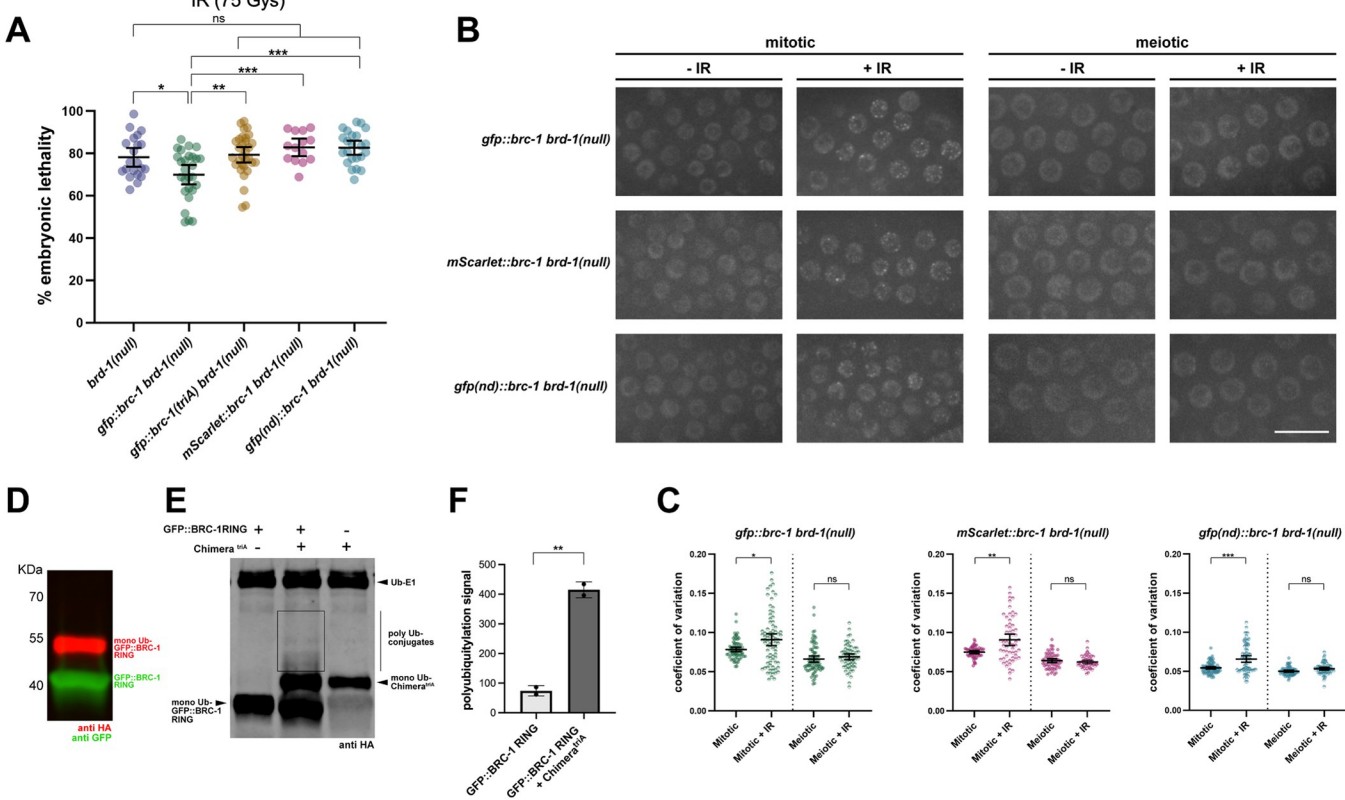

**Fig 7. BRC-1 nuclear accumulation and self-association are required to weakly bypass BRD-1 function.** (A) Embryonic lethality in the presence of 75Gys IR was examined in *brd-1(null)* (n = 21), *gfp::brc-1 brd-1(null)* (n = 27), *gfp::brc-1(triA) brd-1(null)* (n = 31), *mScarlet::brc-1 brd-1(null)* (n = 14), *gfp(nd)::brc-1 brd-1(null)* (n = 23). Mann-Whitney * p < 0.05, ** p < 0.01, *** p < 0.001. (B) Images of live worms of mitotic and meiotic (early-mid pachytene) *brd-1(null)* germ cells in the absence (-IR) and presence of IR (+IR) expressing either GFP::BRC-1, mScarlet::BRC-1 or GFP[nd]::BRC-1. Scale bar = 10μm. (C) Coefficient of variation for GFP::BRC-1, mScarlet::BRC-1, and GFP[nd]::BRC-1 fluorescence in the absence of BRD-1 in the absence and presence of IR in mitotic and meiotic germ cell nuclei; a minimum of 4 germ lines were analyzed for each genotype. Statistical comparisons between—and + IR by Mann-Whitney * p < 0.05, ** p < 0.01, *** p < 0.001. (D) Immunoblot of auto-ubiquitylation (anti-HA-Ub) of GFP::BRC-1 RING; 0.082 +/- 0.025 monoubiquitinylation/total protein. (E) Immunoblot of auto-ubiquitylation of GFP::BRC-1 RING, GFP::BRC-1 RING in the presence of the BRD-1-BRC-1[triA] chimera, and BRD-1-BRC-1[triA] chimera; boxed region shows presence of poly Ub-conjugates. (F) Quantification of polyubiquitylation signal between GFP::BRC-1 RING and GFP::BRC-1 RING in the presence of BRD-1-BRC-1[triA] chimera. p <0.01, Student T test.

was enriched in the nucleus and formed weak foci in response to IR in the absence of BRD-1, although there was reduced steady state levels of GFP::BRC-1 compared to in the presence of BRD-1 (Figs 7B and 7C and S5D and S5E Fig). These findings suggest that GFP::BRC-1 alone can provide some function without its binding partner. Rescue was specific to appending GFP to BRC-1, as neither GFP::BRD-1 nor BRD-1::GFP could provide partial function when BRC-1 was absent as measured by embryonic lethality following IR treatment (S5A Fig). Additionally, neither GFP::BRD-1 nor BRD-1::GFP accumulated in the nucleus in the absence of BRC-1 [42] (S5B Fig).

To determine the consequence of removing BRD-1 to E3 ligase activity we expressed and purified GFP::BRC-1 RING in *E. coli* (S5C Fig) and assayed auto-ubiquitylation *in vitro*. We observed significant auto-monoubiquitylation of GFP::BRC-1 RING (0.082 +/- 0.025 monoubiquitinylation/total protein), but no polyubiquitylation, suggesting that GFP::BRC-1 RING alone is only competent to transfer a single ubiquitin onto itself (Fig 7D). To ascertain whether the lack of polyubiquitylation was due to the absence of BRD-1, we performed the *in vitro* assay using GFP::BRC-1 RING in the presence of the BRD-1-BRC-1$^{triA}$ chimera, which is incapable of polyubiquitylation (Fig 3B). Addition of BRD-1-BRC-1$^{triA}$ to the reaction resulted in both an increase in monoubiquitylation of the chimera (total ubiquitin signal was 2x the BRD-1-BRC-1$^{triA}$ ubiquitin signal alone; Fig 7E) and weak polyubiquitylation (Fig 7E and 7F). These results suggest that BRC-1 can monoubiquitylate itself when fused to GFP in the absence of BRD-1, but that BRD-1 enhances the activity of BRC-1 and is required for polyubiquitylation of the complex.

To determine whether GFP::BRC-1-mediated ubiquitylation was functionally important, we expressed GFP::BRC-1$^{triA}$ in *brd-1(null)* and monitored embryonic lethality following exposure to IR. No significant rescue was observed, suggesting that BRC-1-mediated ubiquitylation is important in response to DNA damage (Fig 7A).

## BRC-1 nuclear accumulation and self-association are important for function in the absence of BRD-1

To ascertain how fusion of GFP to the N-terminus of BRC-1 promotes function in the absence of BRD-1, we constructed an N-terminal fusion with mScarlet, a monomeric red fluorescent protein [65], at the endogenous BRC-1 locus. The mScarlet::BRC-1 fusion was fully functional (S5A Fig). However, expression of mScarlet::BRC-1 in *brd-1(null)* did not improve progeny viability following exposure to IR, even though mScarlet::BRC-1 was nuclear, could form foci in response to IR, and was as stable as GFP::BRC-1 in the *brd-1(null)* mutant (Figs 7A–7C and S5D–S2E). These results suggest that nuclear accumulation, while necessary, is not sufficient for BRC-1 function independent of BRD-1. As the name implies, mScarlet is monomeric, while GFP has the tendency to dimerize or oligomerize, particularly at high concentrations [66]. We next addressed whether association between GFP molecules was important for bypassing BRD-1 function. To that end, we modified the GFP fused to BRC-1 by substituting hydrophobic amino acids with charged amino acids on the surface interface (A206K, L221K, F223R) [67]; we refer to this as GFP$^{nd}$::BRC-1 (nd for non-dimerizable). As with mScarlet::BRC-1, GFP$^{nd}$::BRC-1 is fully functional in an otherwise wild-type worm (S5A Fig). However, expression of GFP$^{nd}$::BRC-1 did not provide any rescue of the *brd-1(null)* mutant even though it was nuclear, could form foci in response to IR, and was as stable as GFP::BRC-1 in the *brd-1(null)* mutant (Figs 7A–7C and S5D–S5E Fig). These results indicate that nuclear accumulation and self-association of BRC-1 driven by GFP can weakly bypass the requirement for BRD-1 in response to DNA damage.

## Discussion

Here we take advantage of *C. elegans* to examine the requirement for BRCA1-BARD1 E3 ubiquitin ligase activity *in vivo* in the context of DNA damage signaling and meiosis. We find that mutants significantly impaired for E3 ligase activity *in vitro* still provide some function *in vivo*. We provide evidence that nuclear localization and BRC-1-BRD-1 association are critical for the function of the complex and these properties are impacted when E3 ligase activity is abrogated. Additionally, we show that GFP fusion to BRC-1 can drive protein accumulation in the nucleus and BRC-1 self-association, which weakly rescues defects in DNA damage repair in the absence of BRD-1, suggesting that BRC-1 is the key functional unit of the complex, while BRD-1 plays an accessory role to augment BRC-1 function.

### A BRCA1 ligase dead mutant?

The role of BRCA1-BARD1 E3 ubiquitin ligase activity has remained enigmatic, due in part to the absence of a true ligase dead allele [19, 31, 32]. Based on extensive biochemical and structural work on RING-type E3 ligases in general, and the human BRCA1-BARD1 complex specifically, we constructed two BRC-1 mutants predicted to interfere with E3 ligase activity: I23A and triA (I23A, I59A, R61A) [14, 18, 19, 68, 69] (Fig 1A and 1B). In human BRCA1 isoleucine 26 defines the binding site for E2 conjugating enzymes, while lysine 65 is the linchpin residue that activates E2-ubiquitin for ubiquitin transfer; BRCA1 harboring the triple I26A, L63A, K65A mutation is an E3 ligase dead mutant *in vitro* [19]. The corresponding isoleucine 23 and arginine 61 residues in *C. elegans* BRC-1 likely play analogous roles in E2 binding and activation of E2-ubiquitin and therefore the triple I23A, I59A, R61A mutant is predicted to be a ligase dead enzyme (Fig 1B). Surprisingly, while both BRC-1$^{I23A}$ and BRC-1$^{triA}$ are significantly impaired for E3 ligase activity *in vitro* (Fig 3), they have different phenotypes *in vivo* (Figs 1, 2 and 4). Further, neither *brc-1(I23A)* nor *brc-1(triA)* has a phenotype as severe as *brc-1(null)*, suggesting that the weak monoubiquitylation observed can promote some E3 ligase function and/or in addition to E3 ligase activity, the complex serves a structural role to promote DNA damage signaling, repair, and meiotic recombination. This latter possibility is consistent with studies of human BRCA1, where RING-less mutants maintain some homologous recombination function [70, 71].

Human BRCA1-BARD1 is capable of coupling with multiple E2s *in vitro* and different E2s define mono vs. polyubiquitylation of substrates and how polyubiquitin chains are linked to each other. The BRCA1$^{I26A}$ mutant has residual E3 ligase activity with a subset of E2s, including UbcH5c, which in complex with BRCA1-BARD1 promotes auto-polyubiquitylation [18, 19]. We used UbcH5c in *in vitro* ubiquitylation assays and observed robust auto-polyubiquitylation as well as mono- and di-ubiquitylation of H2A with the wild-type chimera but no detectable auto-polyubiquitylation with either I23A or triA chimeras, nor any ubiquitylation of histone H2A (Fig 3). However, there was significantly reduced but measurable auto-monoubiquitylation with the mutant chimeras. The residual auto-monoubiquitylation may be only modestly affected by mutation of I59A and R61A in the I23A mutant *in vitro* but may have more profound effects on E3 ligase activity *in vivo*. Further, ubiquitylation assays using different E2s may uncover differences in E3 ligase activity between the I23A and triA mutants that more closely reflect the *in vivo* consequence of these mutations. It is also possible that I59 and R61 do not play analogous roles as to the human protein in terms of E3 ligase activity, although they are required *in vivo*. Alternatively, or in addition, monoubiquitinylation could be a consequence of the RING domains of BRD-1 and BRC-1 being physically tethered in the chimera in our *in vitro* assay. We did observe low levels of monoubiquitylation in the absence of any E2

(S2 Fig), suggesting that some auto-monoubiquitylation results from forced interaction between BRC-1 and BRD-1 RING domains within the chimera.

UbcH5c is orthologous to *C. elegans* UBC-2 (LET-70), which has previously been shown to couple with BRC-1-BRD-1 for ubiquitin transfer in the context of DNA damage signaling [33]. Like the human complex, it is likely that BRC-1-BRD-1 couples with multiple E2s to regulate different pathways (e.g., DNA damage signaling, meiosis, heterochromatin regulation, axon outgrowth) [33, 40–47]. The *C. elegans* genome encodes 22 E2s and the entire spectrum of these E2s coupling to different E3 ligases is not clear [72]. Recently developed tools to conditionally deplete proteins in a tissue-specific manner could help define how different E2s couple with BRC-1-BRD-1, and other E3 ubiquitin ligases, to regulate different pathways *in vivo* [73–75].

While *C. elegans* BRC-1-BRD-1 shares many similarities with the human complex, it is not surprising that differences have evolved between worms and humans. For example, we found that in contrast to the human proteins, *C. elegans* E3 ligase defective BRC-1 showed impaired interaction with BRD-1 (19) (Fig 4). One possibility is that these amino acid substitutions directly alter how BRC-1 and BRD-1 interact, although these do not reside in the helices required for binding between BRCA1 and BARD1. Alternatively, BRC-1-BRD-1 auto-ubiquitylation at specific sites *in vivo* may enhance interaction between these proteins, and these sites are differentially affected by I23A versus triA. It is also possible that using physically tethered BRC-1 and BRD-1 RING domains in our chimeric proteins in the *in vitro* assay may have masked the interaction defect, leading to similar impairment of E3 ligase activity *in vitro* in BRC-1$^{I23A}$ and BRC-1$^{triA}$, but different phenotypes *in vivo*. A recent study has also documented different modes of binding of BRC-1 and BRD-1 with the nucleosome for H2A ubiquitylation compared to the human complex [23]. Nonetheless, continued analyses in *C. elegans* will be instrumental in defining the fundamental roles of BRCA1-BARD1 in the context of a whole organism.

## BARD1 serves an accessory role to ensure BRCA1-mediated polyubiquitylation and nuclear localization

BARD1 was identified as a BRCA1 interacting protein and mutations in BARD1 also lead to an increased incidence of cancer [2, 6, 76]. Structural work defined the contact sites between the two proteins at the helices adjacent to the RING domains and demonstrated that only BRCA1 binds E2s for ubiquitin transfer, while BARD1 is required to stimulate BRCA1's E3 ligase activity [15, 77, 78]. We observed robust auto-polyubiquitylation of the wild-type chimera *in vitro;* however, assaying GFP::BRC-1 RING alone revealed significant auto-monoubiquitylation only, but no polyubiquitylation, in the presence of the same E2. Addition of the triA chimera to the GFP::BRC-1 RING reaction promoted the formation of polyubiquitylation, suggesting that BRD-1 is specifically required for BRC-1-mediated polyubiquitylation. Whether BARD1 also stimulates BRCA1 polyubiquitylation in mammalian cells is unclear; however, it has been shown that BRCA1-BARD1 auto-polyubiquitylation enhances the E3 ligase activity of the full-length complex *in vitro* [14, 15, 79].

In addition to promoting E3 ligase activity, BARD1 is important for the stability and nuclear retention of BRCA1 *in vivo*. Analysis of human BRCA1-BARD1 have revealed multiple mechanisms, including both regulated nuclear import and export driven by interaction between the two proteins, to ensure nuclear localization of the complex where it primarily functions [80, 81]. Similar to what has been reported in mammals, *C. elegans* BRD-1 is required for the stability and nuclear localization of BRC-1 [40, 82]. It was therefore surprising that appending GFP to the N-terminus of BRC-1 could bypass the requirement for BRD-1 in

promoting nuclear accumulation of BRC-1. GFP::BRC-1 alone could weakly promote DNA damage signaling in the absence of BRD-1 and this was dependent on both nuclear localization and self-association driven by GFP. These results reinforce that BRCA1 is the primary functional unit of the complex and its key functions and targets are within the nucleus, while BARD1 serves an accessory role to bolster BRCA1-mediated polyubiquitylation and nuclear localization.

While both BRCA1 and BARD1 possess N-terminal RING and C-terminal BRCT domains, BARD1 uniquely contains conserved ankyrin repeats in the middle of the protein [10]. Recent molecular and structural studies have revealed that the BARD1 ankyrin and BRCT domains direct the interaction of BRCA1-BARD1 to N-terminal ubiquitylated histone H2A within the nucleosome, a chromatin mark associated with DSBs. Once bound, the complex mediates the ubiquitylation of the C-terminal tail of H2A, which opposes the binding of 53BP1 to promote repair by homologous recombination [25–27]. Given the unique requirement for BARD1 ankyrin domains in recruitment to damaged DNA, how does GFP::BRC-1 partially bypass the need for BRD-1 with respect to DNA damage signaling? One possibility is that there are redundant mechanisms for recruitment of BRC-1-BRD-1 to DSBs. Human BRCA1-BARD1 recruitment to DNA damage sites has been shown to be mediated through both BRCA1-BARD1 and RAP80 [83]. While no obvious RAP80 ortholog has been identified in *C. elegans*, other interacting proteins may serve a similar role in the recruitment of the complex to DSBs, and/or sequences within BRC-1 itself may facilitate concentration at DNA damage sites.

### Nucleoplasmic BRC-1-BRD-1 is critical for function

Our analysis of the E3 ligase defective mutants revealed that GFP fusion to BRC-1$^{\text{triA}}$ but not GFP fusion to BRD-1 drives nuclear accumulation. Surprisingly, in response to DNA damage, both GFP::BRC-1$^{\text{triA}}$ and BRD-1::GFP in *brc-1(triA)* can form nuclear foci in proliferating germ cells; however, only GFP::BRC-1$^{\text{triA}}$ is nucleoplasmic in meiotic cells and partially rescues function. Interestingly, BRC-1-BRD-1 foci formation in meiotic cells is dependent on E3 ligase activity. It is unclear why E3 ligase activity plays a differential role in mitosis vs. meiosis in the germ line; however, several features of meiosis provide challenges to DNA repair. Unique to meiosis is the pairing and synapsis of homologous chromosomes, which are essential for crossovers formation to ensure that the homologs segregate properly at Meiosis I. These events occur within the specialized structure of meiotic chromosomes, which includes the chromosome axes and the SC. Chromosome axes are extended filaments, which provide a scaffold for the organization of chromosomes as a linear array of loops [84, 85], and become the lateral elements of the SC. We found that blocking the formation of the chromosome axis, or the SC, did not alleviate the requirement for BRC-1-BRD-1 E3 ligase activity, suggesting that their presence does not impose an additional barrier for recruitment of the complex to meiotic DSBs (Fig 6). In addition to the specialized structure of meiotic chromosomes, the chromatin landscape is also different in meiotic cells and this unique chromatin environment may dictate the requirement for E3 ligase activity in recruiting the complex to meiotic DSBs [86]. Additionally, context-specific BRC-1-BRD-1 post-translational modifications and/or interacting proteins may exist that define redundant pathways for recruiting the complex to DNA damage sites in mitotic germ cells. Future work will provide insight into the context-dependent recruitment of BRC-1-BRD-1 in response to DNA damage.

### Conclusion

BRCA1-BARD1 regulates a plethora of processes *in vivo* and mounting evidence indicates that BRCA1-BARD1 E3 ligase activity is critical for several aspects of the complex's function,

including tumor suppression. However, the underlying molecular mechanisms are just beginning to be revealed. Our findings that BRC-1 is the key driver for DNA damage signaling and repair within the heterodimer is consistent with the observed higher prevalence of pathogenic variants identified in BRCA1 as compared to BARD1 [87, 88]. Further, mutations in BRCA1 pose high risk for both breast and ovarian cancer, while BARD1 mutations are only a risk factor for breast, but not ovarian cancer [89–91]. Thus, as in *C. elegans*, human BRCA1 and BARD1 are not equivalent in function leading to different spectrum of cancers when mutated.

## Materials and methods

### Genetics

*C. elegans* strains used in this study are listed in S1 Table. Some nematode strains were provided by the Caenorhabditis Genetics Center, which is funded by the National Institutes of Health National Center for Research Resources (NIH NCRR). Strains were maintained at 20˚C.

### CRISPR-mediated allele construction

*brc-1(xoe4)*, *gfp::brc-1(xoe7)* and *brd-1::gfp(xoe14)* have been described [42]. *brd-1(xoe58[null-gfp::3xFLAG])* used the same guide and template as *brd-1::gfp(xoe14)* [42]. *gfp::brc-1(xoe20 [I23A])* and *mScarlet-i::brc-1(xoe34)* were generated using CRISPR-mediated genome editing with a self-excising cassette as described in [92] with modifications as follows: I23A was introduced at the same time with GFP knock-in by incorporating the corresponding mutation in the 3' homology arm on the repair template plasmid using the Q5 site-directed mutagenesis kit (New England Biolabs). GermLine Optimized mscarlet-i sequence [82] was cloned into the repair template plasmid in place of GFP by Gibson Assembly to generate *mscarlet-i::brc-1 (xoe34)*. *gfp::brc-1(xoe48[I23A, I59A, R61A])* was generated by introducing the corresponding I59A R61A mutations in the *gfp::brc-1(xoe20[I23A])* background using the co-CRISPR method [93]. All other genome-edited strains were generated using the co-CRISPR method. Guide sequence, repair template, and primers for genotyping are provided in S2 Table. All strains were outcrossed for a minimum of three times before analyses.

### Embryonic lethality in the absence and presence of irradiation and male self-progeny

L4 hermaphrodites were transferred to individual plates (-IR) or exposed to 75Gys γ-irradiation from a $^{137}$Cs source, and then transferred to individual plates. The resulting hermaphrodites were transferred to new plates every 24hr for 3 days. Embryonic lethality was determined by counting eggs and hatched larvae 24hr after removing the hermaphrodite and percent was calculated as eggs/(eggs + larvae). Males were scored after 72hr and calculating percent as males/(males + hermaphrodites + eggs).

**Cytological analyses.** Immunolabeling of germ lines was performed as described [94]. The following primary antibodies were used at the indicated dilutions: rabbit anti-RAD-51 (2948.00.02; SDIX; 1:10,000; RRID: AB_2616441), rabbit anti-BRD-1 (1:500; from Dr. Simon Boulton [33]), rabbit anti-SYP-2 (1:250; from Dr. Sarit Smolikove). Secondary antibodies Alexa Fluor 594 donkey anti-rabbit IgG, and Alexa Fluor 488 goat anti-mouse IgG from Life Technologies were used at 1:500 dilutions. DAPI (2µg/ml; Sigma-Aldrich) was used to counterstain DNA.

Collection of fixed images was performed using an API Delta Vision Ultra deconvolution microscope equipped with an 60x, NA 1.49 objective lens, and appropriate filters for epi-

fluorescence. Z-stacks (0.2μm) were collected from the entire gonad. Images were deconvolved using Applied Precision SoftWoRx batch deconvolution software and subsequently processed and analyzed using Fiji (ImageJ) (Wayne Rasband, NIH).

RAD-51 foci were quantified in a minimum of three germ lines of age-matched hermaphrodites (18–24 hr post-L4). As the *zim-1* mutation results in an extended transition zone, we divided germ lines into four equal zones beginning from the first row with two or more crescent-shaped nuclei until the end of pachytene (Fig 2B). RAD-51 foci were quantified from half projections of the germ lines; the number of foci per nucleus was scored for each region.

To measure pixel intensities of RAD-51, foci were identified by a prominence value between 10–20 using the "Find Maxima" function embedded in Fiji from half projections of germ lines. Pixel intensities were measured from defined regions of the gonad and were plotted with their means and 95% CI using GraphPad Prism. A minimum of three germ lines for each genotype were used for quantification.

The nuclear to cytoplasmic ratio of mitotic and meiotic (early to mid pachytene) regions of the gonad was measured by determining nuclear and cytoplasmic mean pixel intensity of BRD-1 immunolabeling or direct GFP fluorescence signal using Fiji. The nucleoplasmic to cytoplasmic ratio was calculated for each individual nucleus and surrounding cytoplasm; multiple values were pooled together for a specific region as indicated. The mean and 95% CI for the ratio was plotted. A minimum of three germ lines were analyzed.

For live cell imaging, 18–24 hr post L4 hermaphrodites were anesthetized in 1mM tetramisole and immobilized between a coverslip and a 2% agarose pad on a glass slide. Z-stacks (0.33μm) were captured on a spinning-disk module of an inverted objective fluorescence microscope with a ~100Å, NA 1.46 objective, and EMCCD camera. Z-projections of stacks were generated, cropped, and adjusted for brightness in Fiji.

GFP::3xFLAG::BRC-1$^{triA}$ fluorescence following exposure to 75Gys IR was quantified by measuring the mean fluorescence intensity and standard deviation (SD) in Fiji for individual nuclei [region of interest (ROI)] in mitotic germ cells (proliferative zone) and meiotic germ cells (early to mid pachytene). Coefficient of variation (CV) is defined as SD of intensity divided by mean intensity [55]. The CV describes the dispersion of pixel intensity values from a 2D ROI around the mean pixel intensity such that nuclei with more distinct foci will have high CV values, whereas nuclei with more uniform fluorescence will have low CV values.

GFP::BRC-1, and GFP::BRC-1$^{triA}$ foci were quantified in 10 mid-pachytene nuclei from each of three half projections of germ lines of age-matched hermaphrodites (18–24 hr post-L4).

**Protein constructs.** The BRD-1-BRC-1 chimera, encoding amino acids 1–107 of BRD-1 and amino acids 2–106 of BRC-1 separated by a GGSGG-linker was synthesized as a G-block and cloned into pET28A vector containing a PreScission protease cleavage site, sfGFP and a strepII-tag, using Gibson Assembly. Mutant BRD-1-BRC-1 chimeras harboring the single I23A mutation and the I23A I59A R61A triple point mutations were similarly synthesized as G-blocks and cloned into pET28A as described above. The GFP::BRC-1 RING construct in pET28A encodes amino acids 2–106 of BRC-1 and a N-terminal GFP followed by 3x FLAG-tag and a C-terminal strepII-tag. This construct contains identical amino acid sequences as the fusion protein expressed in *gfp::brc-1(xoe7)* allele with the exception of truncated BRC-1 RING domain and the addition of the strepII-tag.

**Protein purification.** The wild-type and mutant BRD-1-BRC-1 RING chimeras were expressed in BL21-CodonPlus (DE3)-RIPL cells (Agilent). The cells were grown at 37˚C until OD600 0.6 and were induced by 0.2mM isopropyl-β-D-thiogalactoside in the presence of 100μM ZnCl2 at 37˚C for 6 hrs. The GFP::BRC-1 RING was induced overnight at 18˚C. After induction, cells were harvested and resuspended in buffer A (20mM HEPES-KOH pH 7.2,

300mM NaCl, 1mM EGTA) supplemented with 1mM DTT, 0.2% NP-40, protease inhibitors (1mM PMSF; protease inhibitor cocktail P83340; Sigma-Aldrich) and lysed using a Emulsiflex C-3 (Avestin) high pressure homogenizer. The lysates were centrifuged at 15000xg for 20min at 4˚C. The supernatants were passed through Strep-Tactin XT (IBA) for affinity purification, and the column was washed with lysis buffer to remove unbound proteins before eluting the proteins with 50 μM biotin (Chem-Impex Int'l) in low salt buffer (20mM HEPES-KOH pH 7.2, 30 mM NaCl, 0.1% NP40). Proteins were further purified by anion exchange using HiTrap Q HP column equilibrated with 20mM HEPES-KOH pH7.2 with a linear NaCl gradient from 0mM to 600mM. Peak fractions were pooled and concentrated on Amicon-Ultra spin filters (EMD Millipore) and supplemented with 10% glycerol. Protein aliquots were snap frozen in liquid nitrogen and stored at -80˚C. Protein concentration was measured using a Nanodrop One (ThermoFisher) based on the total amount of fluorophore (sfGFP or GFP).

**E3 ligase activity assay.** Ubiquitin transfer reactions were performed in 30μl reaction volume at 30˚C for the indicated time with mild rocking. For end point auto-ubiquitylation assays, the reaction mixture contained 0.2μM E1 (hUBE1; E-305; biotechne), 1uM E2 (hUbcH5c; E2-627; biotechne), 5μM BRD-1-BRC-1 chimera, 20μM HA-ubiquitin (U-110; bitechne), 5mM ATP, 5mM MgCl$_2$ in reaction buffer (20mM Hepes pH 7.2; 150mM NaCl). To test ubiquitylation of histone H2A, 0.75μM human histone H2A (ab200295; Abcam) was added to the above reaction mixture. For time course experiments, 0.1μM E1, 0.5μM E2, 3μM E3 chimera, 10μM HA-ubiquitin, 5mM ATP, 5mM MgCl$_2$ were mixed in a 150μl reaction volume and incubated at 30˚C. 30μl were removed at 0, 5, 10, 20, 40min, the reaction stopped with 10μl 4X sample buffer and boiled. Reaction mixtures were visualized by immunoblot and analyzed by measuring pixel intensity of ubiquitylated species.

**Immunoblot analysis.** Whole worm lysates were generated from indicated worms. ~100 worms were collected, washed in M9 buffer and resuspended in equal volume of 2X Laemmli sample buffer (Bio-RAD). Worm lysates or E3 ligase reaction mixtures were resolved on 4–20% stain-free SDS-PAGE gels (Bio-RAD) and transferred to Millipore Immobilon-P PVDF membranes. Membranes were blocked with 5% nonfat milk and probed with mouse anti-FLAG (MA1-91878; Invitrogen; 1:1000; RRID AB_1957945), rabbit anti-GFP (NB600-308; Novus Biologicals; 1:2000; RRID: AB_10003058), mouse anti-HA [12CA5; amino acids 98–106 of human influenza virus hemagglutinin protein; IgG2b mAb; 1:1000; RRID: AB_2532070; in-house (Trimmer Laboratory)], or anti-Histone-H2A (ab18255; Abcam; 1:1000; RRID:AB_470265) followed by IRDye800-conjugated anti-mouse IgG secondary antibodies (962 32212; LI-COR Bioscience; 1:20000; RRID: AB_621847). Immunoblots were imaged on a LI-COR Odyssey Infrared Imager, signal was quantified using Fiji and normalized with total protein stain.

**Yeast 2 hybrid.** Full length wild-type or mutant BRC-1 sequences were cloned into plasmid pBridge (Takara Bio), transformed into yeast strain Y2HGold (Takara Bio) and plasmids were selected on medium lacking tryptophan. Full length BRD-1 sequences were cloned into plasmid pACT2.2, transformed into yeast strain Y187 (Takara Bio) and transformants were selected on medium lacking leucine. Wild type or mutant BRC-1 expressing strains were mated with BRD-1 expressing strain and the diploids selected on -Trp-Leu double drop out plate at 30C. Diploid cells were grown in liquid -Trp-Leu medium overnight, and serial dilutions were plated on -His -Trp -Leu and -Trp -Leu solid media. For quantitative measurement of wild type or mutant BRC-1 and BRD-1 interactions, supernatants from liquid cultures were assayed using CPRG (chlorophenol red-b-D-galactopyranoside, RocheApplied Science Cat. NO.10884308001) as substrate and β-galactosidase units were calculated as described (Yeast Protocol Handbook, Takara Bio, https://www.takarabio.com/products/protein-research/two-hybrid-and-one-hybrid-systems/yeast-two-hybrid-system/matchmaker-gold-yeast-two-hybrid-system).

**RT-PCR.** Total RNA was isolated from 50 to 100 μl of packed worms from wild type and *brd-1(null)* using the RNeasy Mini Kit (74104; Qiagen) and QIAshredder (79654; Qiagen). One μg of RNA was converted to cDNA using SuperScript III First-Strand Synthesis System for RT-PCR (18080–051; Invitrogen) primed with Oligo (dT)20. PCR was performed in a standard PCR machine using 20 cycles.

## Supporting information

**S1 Table. Strains.**
(DOCX)

**S2 Table. CRISPR alleles.**
(DOCX)

**S3 Table. RAD-51 statistical analyses.**
(DOCX)

**S1 Fig. *brc-1(I59A, R61A)* is phenotypically wild type.** (A) Embryonic lethality (left Y axis) and male self-progeny (right Y axis) of wild type and *brc-1(I59A, R61A)* worms; n = 12, except for wild type embryonic lethality, n = 26. (B) Embryonic lethality in the presence of 75Gys IR of wild type (n = 36) and *brc-1(I59A, R61A)* (n = 23) worms. ns = not significant Mann-Whitney.
(TIF)

**S2 Fig. BRD-1-BRC-1 chimera protein purification and E3 ubiquitin ligase assays.** (A) Purified chimera proteins visualized on stain-free gels (proteins do not run true to size as they were loaded on gel in sample buffer without heat denaturation) with indicated molecular weight markers in kDaltons. (B) Titration of E2 conjugating enzyme in E3 ubiquitin ligase assay shows a non-specific mono-ub conjugate product in the absence of E2 enzyme. (C) Incorporation of mono- and di-HA-ubiquitin into histone H2A as visualized by antibody against HA.
(TIF)

**S3 Fig. BRD-1::GFP in *brc-1(triA)* and BRC-1$^{triA}$::HA are not nuclear nor rescue embryonic lethality in response to IR treatment.** (A) BRD-1 protein localization shown by direct GFP fluorescence in live wild-type and mutant *brc-1* worms in respective germ line regions. PZ = proliferative zone; TZ = transition zone; EP = early pachytene; MP = mid pachytene; LP = late pachytene; DP = diplotene. Scale bar = 10μm. (B) Immunostaining of BRC-1::HA and BRC-1$^{triA}$::HA (red in merge) in fixed mitotic and meiotic (mid-pachytene) nuclei counterstained with DAPI (blue in merge). Scale bar = 10μm. (C) BRD-1::GFP and *brc-1(triA)* BRD-1::GFP fluorescence in live worms in mitotic and meiotic regions (mid-pachytene) in the presence of 75Gys IR. (D) GFP::BRC-1 and GFP::BRC-1$^{triA}$ fluorescence in live worms at late pachytene (LP) showing association with SC and crossover sites. (E) Embryonic lethality of worms treated with 75Gys IR. C-terminal GFP fusion to BRD-1 did not rescue viability in the *brc-1* mutants. (F) Immunoblot (left) showing steady state levels of BRC-1 proteins from wild-type and mutant whole worm extracts. Levels of mutant BRC-1 proteins normalized to wild type protein from three independent experiments (right). ** p<0.01.
(TIF)

**S4 Fig. Analysis of *brd-1(null)*.** (A) BRD-1 exon structure and position of insertion of the stopin cassette. Primer pairs (P1-P3) used for RT-PCR of wild type and *brd-1(null)* cDNA are indicated. P1 Forward: cgccacatttcaacagaaacc, P1 Reverse: gcttctttgctgtagtcgtg; P2 Forward:

cgcgtaattcgacaaaacgc, P2 Reverse: gcattaataactgcacccgc; P3 Forward: ggctcaacattagaaacaacgc, P3 Reverse: gatcaataatgcacgctctcag. *ama-1* was used as control [95]. (B) Immunoblot of whole worm extracts of BRD-1::GFP::3xFLAG and BRD-1[null]::GFP::3xFLAG with indicated molecular weight markers. (C) No GFP fluorescence was observed in *brd-1(null)::gfp* worms. Scale bar = 10μm. (D) Male self-progeny (left Y axis; n = 12) and embryonic lethality (right Y axis; n = 18) of *brc-1(null)* and *brd-1(null)* worms. No statistical differences were observed by Mann-Whitney.
(TIF)

**S5 Fig. Specificity of rescue of GFP::BRC-1 and GFP::BRC-1 RING purification.** (A) Embryonic lethality in the presence of 75Gys IR was examined in *brd-1(null)* (n = 21), *brc-1 (null)* (n = 21), *gfp::brd-1* (n = 12), *brc-1(null) gfp::brd-1* (n = 10), *brd-1::gfp* (n = 11), *brc-1 (null) brd-1::gfp* (n = 18), *mScarlet::brc-1* (n = 14), *gfp(nd)::brc-1* (n = 14). (B) Image of GFP:: BRD-1 fluorescence in the *brc-1(null)* mutant. Scale bar = 5μm. (C) *E. coli* purified GFP::BRC-1 RING protein visualized on a stain-free gel with indicated molecular weight markers. (D) Immunoblot of GFP::BRC-1, mScarlet::BRC-1, and GFP[nd]::BRC-1 in the *brd-1(null)* mutant. (E) Quantification of relative steady state levels of GFP::BRC-1, mScarlet::BRC-1, and GFP[nd]:: BRC-1 in the *brd-1(null)* mutant.
(TIF)

## Acknowledgments

We thank the Caenorhabditis Genetic Center, which is funded by NIH Office of Research Infrastructure Programs (P40 OD010440) for providing strains, Dr. Nicola Silva (Masaryk University) for the *brc-1*::*HA* allele, Dr. Sarit Smolikove (University of Iowa) for the anti-SYP-2 antibody, and Dr. Simon Bolton (The Francis Crick Institute) for the anti-BRD-1 antibody. We also thank Dr. Satoshi Namekawa (University of California Davis) for the histone H2A antibody and Brian Wong for constructing strains. We are particularly grateful to Dr. Judy Callis (University of California Davis) for input on E3 ligase activity assays, Dr. Daniel Elatan (University of California Davis) for help with AlphaFold and ChimeraX as well as imaging analyses and the Engebrecht lab for thoughtful discussions. We thank the MCB Light Microscopy Imaging Facility, which is a UC Davis Campus Core Research Facility, for the use of the Deltavision Ultra and 3i Spinning Disc microscopes for generating images.

## Author Contributions

**Conceptualization:** Qianyan Li, JoAnne Engebrecht.

**Data curation:** Qianyan Li.

**Formal analysis:** Qianyan Li, JoAnne Engebrecht.

**Funding acquisition:** JoAnne Engebrecht.

**Investigation:** Qianyan Li, Arshdeep Kaur, Kyoko Okada, JoAnne Engebrecht.

**Methodology:** Qianyan Li, Kyoko Okada, Richard J. McKenney, JoAnne Engebrecht.

**Project administration:** JoAnne Engebrecht.

**Resources:** Richard J. McKenney.

**Supervision:** Qianyan Li, Richard J. McKenney, JoAnne Engebrecht.

**Validation:** Arshdeep Kaur.

**Visualization:** Qianyan Li, JoAnne Engebrecht.

**Writing – original draft:** Qianyan Li, JoAnne Engebrecht.

**Writing – review & editing:** Arshdeep Kaur, Kyoko Okada, Richard J. McKenney.

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
