## [Decision Letter · Decision Letter 0]

9 Nov 2022

Dear Dr Engebrecht,

Thank you very much for submitting your Research Article entitled 'Differential requirement for BRCA1-BARD1 E3 ubiquitin ligase activity in DNA damage repair and meiosis in the Caenorhabditis elegans germ line' to PLOS Genetics.

The manuscript was fully evaluated at the editorial level and by independent peer reviewers. The reviewers appreciated the attention to an important topic but identified some concerns that we ask you address in a revised manuscript. These concerns can be mostly addressed by changes to the text and figures, as well as by inclusion of additional statistical analysis and quantification of existing data. While analysis of BRC-1 localization in the mutants would certainly enrich the conclusions, it is not essential for this revision.

We therefore ask you to modify the manuscript according to the review recommendations. Your revisions should address the specific points made by each reviewer.

Yours sincerely,

Mónica P. Colaiácovo

Academic Editor

PLOS Genetics

Gregory P. Copenhaver

Editor-in-Chief

PLOS Genetics

Reviewer's Responses to Questions

**Comments to the Authors:**

Reviewer #1: In this new work by the Engebrecht lab, the authors explore the functional requirements imposed by the E3 ubiquitin ligase activity of the BRCA1-BARD1 heterodimer during C. elegans meiosis.

Through genetic, biochemical and cytological assays, Li and colleagues show that mutations in different conserved residues within BRCA1's RING domain, while abrogating the ubiquitin ligase enzymatic function in a comparable fashion, yet they confer different phenotypes in vivo.

In particular, they show that the I23A mutation is sufficient, on its own, to block ubiquitin ligase activity but it does not trigger dramatic effects during meiosis, while combining this mutation with I59A R61A (called triA) elicits more severe effects, causing highly similar, although not completely identical phenotypes to brc-1 null worms.

Furthermore, the authors show that nuclear recruitment of BRC-1triA improves viability after DNA damage, indicating partial roles exerted by presence of the protein in the nucleus, and interestingly, they identify a more prominent role for BRC-1 than BRD-1 in performing the heterodimer’s functions. This highlights crucial, pathological implications also for BRCA1-BARD1 in humans, since mutations found in BRCA1-mediated cancer prone syndromes occur with an overwhelmingly higher frequency compared to those in BARD1.

This is beautiful work, which significantly expands our knowledge on the functions exerted by this essential heterodimer in an in vivo system, and as such, it provides invaluable details on the biological relevance of the E3 ubiquitin ligase activity carried out by the complex, an aspect that has been previously studied in other systems but that, however, has produced conflicting and often unconvincing results. The authors have done an excellent job at designing the experimental setting, and the quality of their data is undoubtedly of the highest standards. I have a few issues that should be addressed before publication in PLoS Genetics.

1. The fact that the I23A single mutation and the triA have synthetic phenotypic effects while having nearly identical consequences on abrogating the E3 ubiquitin ligase activity, highlights essential roles for the ubiquitin ligase activity carried out individually by the two aminoacidic modules but possibly underlines other redundant/separate functions in vivo. This may also suggest that while lack of ubiquitin ligase function does not dramatically impair, by itself, worms viability (as in the I23A mutant), the combination with the I59A R61A mutations could block other functions both before and after ectopic DNA damage. Given the high proficiency of the authors with CrispR and the fact that it should be quite straightforward to mutate these two residues in the clones that they have already expressed in E. coli, I would like to see what are the phenotypical consequences of mutating the I59A R61A while leaving untouched I23.

Is BRC-1I59A R61A equally impaired in the ubiquitin ligase activity?

Are the brc-1I59A R61A mutant worms similarly sterile, before and after IR, to the I23A or to the triA mutants?

2. It would be interesting to know whether lack of ligase activity impacts not only RAD-51 dynamics but also CO establishment. Have the authors assessed COSA-1 foci formation and DAPI-stained bodies in the brc-1triA; zim-1 vs the brc-1(null); zim-1?

3. In Fig. 4A, BRD-1 does not really look like “short stretches” to me, but rather foci in both the I23A and the triA mutants.

4. Lines 256-260. I would be more cautious in giving as a fact that BRD-1 interaction with mutated BRC-1 is impaired. The Y2H data is solid, but since this is not an in vivo assay, the authors should tone down their conclusions.

5. Lines 333-341. The authors did well in generating a novel putative null allele of brd-1, which indeed behaves as expected. However, given that the editing strategy involved insertion of multiple STOP codon rather than a full deletion, the authors should provide a Western Blot probed with the anti-BRD-1 antibody in their possession to show no BRD-1 detection in their novel brd-1 allele. The transcript is indeed destabilized, however, especially with the P3 oligos, there is still substantial amplification (Fig. S3A). There is no GFP detection by live imaging, however this is not definitive proof that the protein might be below detection levels. As a side note, I would like to mention that for two of the available brd-1 deletion alleles mentioned by the authors, the brd-1(dw1) and brd-1(gk297), it was shown that they are both null mutants, as no detectable BRD-1 by Western Blot was observed (Janisiw et al.; 2018, Fig. S1C).

6. Lines 342-343. Although the small difference in the embryos viability is indeed (weakly) statistically significant, “partial rescue” sounds slightly overemphasized. Please tone down.

7. Please indicate in Fig. S4A that viability assessment was performed upon IR exposure.

8. Fig. 7. Please include a panel with the GFP::brc-1 (brd-1 +/+) in Fig. 7B. Further, the assay shown in Fig. 7D is not entirely clear to me: do the authors observe monoubiquitylation by observing the intensity of the band? Except for that, the pattern between the middle and last lanes seems the same to me.

9. Also, I would like to see the Western Blot quantified in Fig. S4D and please include more details in Fig. S4C, as there is no indication to what we are looking at in the figure.

10. Given that different effects have been observed in the rescue conferred by the GFP fusion, it would be highly relevant to assess whether this phenotype is genuinely due to the nuclear recruitment or rather caused by independent, stabilizing effects of the GFP on the mutated protein. To this end, the authors should monitor the nuclear recruitment and the phenotypical effects on a differently tagged version of brc-1 without fluorescently-labelled tags (HA, FLAG, etc…). There is a brc-1::HA functional tagged strain available in the community (Janisiw et al.; 2018), however I do realize that in this case, the position of the tag is at a different end compared to the strain used by the authors (GFP::brc-1), which could trigger unforeseen effects. The authors could either generate the triA mutation in the brc-1::HA anyway or excise the GFP tag from the triA mutant that they already have, leaving only the 3xFLAG.

11. Please indicate the molecular weight on the bands of the ladder in Fig. S1A-C, Fig. 4C, Fig. S2C, Fig. S4C and Fig. 7D.

12. Please indicate the genotype in all the charts: the colour code is helpful but it would still be more practical to have the genotypes specified in each chart. Specifically in Figs. 2B, E and F.

Reviewer #2: The manuscript by Li et al identifies meiotic roles for BRC-1-mediated poly-ubiquitination in the stabilization of the BRD-1/BRC-1 complex and therefore in the downstream events regulating recombination. The manuscript uses a combination of cell biology and genetics approaches and make an exceptionally appreciated effort to address the biochemical activity of their mutants. The paper reports an interesting finding: removing all polyubiquitination activity and >90% of mono-ubiquitination activity of BRC-1 results in mild phenotypic consequences. This finding therefore suggests that the meiotic functions of BRC-1 may be dependent on weak mono-ubiquitination activity or are unrelated to its function as a ubiquitin ligase (possibly reflecting a structural role). Either one of these options is interesting for the field and for future identification of meiotic BRC-1/BRD-1 substrates. The finding provide evidence that that poly-ubiquitination (gone via I23A) plays a role in stabilizing the BRC-1/BRD-1 complex and further destabilization of this complex (tri mutant) is responsible for the downstream meiosis/DSB repair phenotypes. The authors make a reasonable argument that dimerization of BRC-1 (via GFP fusion) can bypass some of the defects found in the mutants, suggesting that forcing BRC-1 homodimers may harbor activity similar to the BRC-1-BRD-1 heterodimer. This is another interesting finding. While the studies are well done and presented clearly, some adjustments need to be made with special attention to adding the missing statistical analyses, several experimental control, and changes the language in some places to make more restricted conclusion.

Major comments:

• Introduction: given that this paper focuses on C. elegans, adding a paragraph summarizing the phenotypic consequence of deleting brc-1/brd-1 in C. elegans is important

• Introduction: one of the major effects of the mutant analyzed is that they abrogate poly but not mono ubiquitination of BRCA1. To make the reader prepared for reading the results section, the introduction needs to be more specific about what is known about the roles of mono- vs. poly ubiquitination and the function of autoubiquitination of BRCA1 throughout systems.

• It is very confusing to start looking at the phenotypes before validating that the mutants lack ubiquitination. The results section will be easier to read if Figure 3 would come first, with panel A and B from current figure 1 moved to figure 3 (which would be the new figure 1)

• The concluding sentences in each section connect E3 ligase activity to a phenotype, but this is mainly based on the tri mutans that may have two effects: one on poly-Ubiquitination and another on complex stability in ubiquitination unrelated manner. While its reasonable that ubiquitination plays a role, it not necessarily a major role. The authors favor the hypothesis that I23A and tri mutants show severalty in ubiquitination phenotype that is under detected by their biochemical assay (line 427-430). However, the data presented is more consistent with I59A and R61A perturbing the structure but not the biochemical function of the complex. While it is reasonable that the I23A mutation links BRC-1/BRD-1complex stability with ubiquitination, the additional 2 mutations that are found in the triA mutant may destabilize the complex in a manner unrelated to the ubiquitin ligase activity. The fact that dimerizable GFP can suppress the phenotypes of the triA mutant also suggests that part of the defects observed in the triA mutant may be attributed to perturbing protein-protein interactions.

• Related: since the mutants do not affect mono-ubiquitination, it is confusion that in the concluding sentences of each section the mutants are described as lacking ubiquitination activity all together when they may just lack poly-ubiquitination activity. It is indeed possible that the chemotic assay creates conditions that are not applicable in vivo (line 427-430), but this was not demonstrated, so we need to assume mono-ubiquitination is still present. It will be better to reword these sentences so they won’t make commitment to the mutants lacking any ubiquitin ligase activity.

• The authors do not analyze the localization pattern of BRC-1 in their mutants since it is mutually exclusive with BRD-1 (using null) and they do examine BRD-1. However, since they are analyzing hylomorphic alleles, it is possible that BRC-1 and BRD-1 localization may not be identical in these mutants. Indeed, based on figure 5B, it may be resendable to expect that the localization of BRC-1 (with no GFP) in the mutants will be different than that of BRD-1.

• Line 271-272 and Figure 5B- it is not clear from the description or image (just early pachytene - mid pachytene is showed) what is the phenotype of gfp::brc-1(triA) in term of localization in LP. Is it just nuclear localization throughout the germline with no association to SC/CO sites or do they observe association to CO sites (like for BRD-1 in brc-1(triA), figure 4A and S2)?

• Line 283-283 claims that the GFP fusion leads to nuclear localization in response to IR, bust since we don’t know how brc-1(triA) localization is altered by IR this cannot be concluded. How brc-1(triA) is affected by IR?

• Line 283-283 what does “intendedly recruited” means? If this meaning to say BRD-1 is not recruited in response to IR in gfp::brc-1(triA) that should be shown in this figure.

• Figure 6ABC should show gfp::brc-1(triA) and gfp::brc-1(triA) side by side for comparison and images from MP-LP should be shown for all genotypes, since it’s not only the foci but also the linear localization patten that is of interest here. For the cohesion mutants an image with no IR is needed as well.

• Line 321 and figure 6ABC- claims for PC formation should be addressed by co-staining BRC-1 with SC SYP protein

• Fig 2BCEF, 3E, 4BDF, 5C, 6D, S2C, S4AD are the differences statistically significant? Based on how the data looks I assume the conclusions of the authors are right, but this should be backed up by statistics.

Other comments

Text:

• Line 69-73 and 207 add reference showing that C. elegans BRC-1/BRD-1 ubiquitinates H2B (36250637)

• Line 123, the authors should explain why they chose to mutate just I23 (as opposed to just 59 or 61). It can be assumed that it stems from the work in mice, indicated in the introduction, but it will be good to be specific about the motivation here to do particular mutant combinations and not others in one sentence or so.

• Line 136 “important when DNA damage is present” to “important when exogenous DNA damage is present”

• Line 160. Please define what stages of meiosis the zones are

• Lin 179-184, this statement is premature here since ligase activity and complex integrity was not examined yet (this may be different if the order of figures will be reorganized as suggested).

• Line 203-204, I propose to remove the sentence “While it did not reach statistical significance, the triA chimera showed consistently lower auto-monoubiquitylation than the I23A chimera” if it’s not statistically different it should not be stated to avoid confusion

• Line 263-266 it will be helpful to mention here that BRC-1 BRD-1 interact though the N’

• Line 299, more severe compared to what?

• Line 287 title “BRC-1-BRD-1 E3 ligase activity is essential for recruitment of the complex to meiotic DSBs” and concluding sentence line 329-330. The rational for this section needs to be explained in more detail. The experiments in figure 5B clearly show that BRC-1 is not recruited to SPO-11 or IR induced DSBs, therefore the results in figure 6A are not surprising. They merely describe a different manner in which meiotic or IR induced DSBs are prolonged by impairing their efficient repair.

• line 299-231 (also related to Line 508-510) The rational provided in line 299-231 is not clear because if anything the SC is promoting BRC-1 recruitment (Janisiw et al 2018 and Li at al 2018). I understand the difference between axis and central region analysis, but not what motivated looking into the effect of axis removal.

• Line 332 and on describing figure 7. It’s not clear what is the motivation for creating brd-1(null) since in Li at al 2018 the authors show that an existing truncation alleles already lacks wt BRC-1 localization, therefore it’s not surprising that a null will lack this localization…

• Line 342-343: it’s hard to make a conclusion here without plotting on the same graph

Discussion

• Can the authors provide an explanation to why the triA mutants still have auto mono-ubiquitination?

• Can the authors comment about why the dark zone size changes in brc-1(I23A)?

• Line 415-417: since mono-ubiquitination is not abrogated there is always a reason to think that is why it’s not a null

Figures

• Fig 2A, 5A. 6E 7A It will be easier to spell out “emb” as “embryonic lethality” for readers outside the C. elegans field.

• Fig 2A, 4B 5C- Y axis should start at 0

• Fig 3C, 6E 7A-some comparisons are missing

• Fig 4A – the DAPI is hardly visible – can it be presented as B&W?

• Fig 5- since figure 5 has many panels with IR, please specify that C is without IR, to make it easy for the reader.

• Fig 6D and Fig 7C Please note which regions of the germline measurements for meiotic nuclei are done in the figure legend and/or figure (since the meiotic region is very large and diverse)

• Fig 7A- please add gfp::brc-1(triA) and brc-1(null) to the graphs since comparisons are made in the text. Also add in the title of the panel that it is 75Gy IR.

• Fig 7D- poly Ubiquitination in the ++ lane is not observable and the quantification of the signals is not shown

• Fig S4A- Please indicate the IR treatment in the graph title

• DAPI channels are lacking in all images that use GFP, assuming because these are taken live. It will be nice to show fixed samples co-stained with DAPI, but if this is not the preference, indicate in the figure legend that these were live imaged.

Typos

• Line 66- “mechanistic insights”

• Line 556- “…and percent was calculated as”

• Line 567- “Z-stacks (0.2�m)…”

Reviewer #3: Review Plos Genetics Li et al 2022

The in vivo role of BRCA1 E3 ligase activity and the specific roles of BRCA1 and BARD1 are not well understood. Partially because a complete null for BRCA1 E3 ligase activity has not been examined in vivo. The authors generated single (I123A) and triple (I23A, I59A, R61A) mutations that should partially and completely abrogate E3 ligase activity respectively in BRC-1. They then examined the roles of these mutant proteins in DNA damage repair and meiotic progression. They find the triA had a more severe phenotype than I123A, for elevated levels of male self progeny and embryonic lethality. Although less severe than the BRC-1 null. IR exposure showed the single mutant had higher embryonic lethality than WT, and the triple mutant was similar to null suggesting E3 ligase activity is critical for DNA damage repair. When meiosis is challenged by perturbing homolog pairing (zim-1 mutation) embryonic lethality was progressively increased in I123A < TriA < null. BRC-1 plays an important role to stabilize RAD-51 nucleoprotein filaments. The authors find that a milder phenotype in RAD-51 foci dynamics in the I123A, zim-1 while the the triple looks similar to null. In terms of intensity all three mutants were similar. In vitro, the E3 ubiquitin ligase activity was equally impaired between I123A and TriA providing little help with the differences in vivo. Intriguingly, nuclear accumulation of the obligate partner BRD-1 was much reduced in the I123A and even further in the triple mutant. The reduction is not reflected in the steady state levels which are only slightly reduced and similar between the mutants. However, in vitro there is a reduction in BRC-1 mutant interaction with BRD-1, in contrast to the human proteins. Intriguingly, fusion of GFP to BRC-1 partially rescues the phenotype of the triple mutant in mitotic but not meiotic cells. The authors measured the distribution of the fusion proteins with and without IR to show that the GFP::BRC-1 (triA) has higher nuclear localization than BRC-1 (triA) suggesting that independently enriching for BRC-1 in the nucleus can improve DNA damage repair in mitotic cells. To further investigate why the meiotic cells lacked BRC-1 (triA) foci, they analyzed the phenotype in worms lacking syp-1 (an SC central element protein), him-3 (meiotic chromosome axis formation), or meiosis specific kleisins in the presence of IR. While GFP::Brc-1 accumulates as foci when synapsis fails, or meiotic chromosome axes or cohesion are disrupted, GFP::Brc-1 (TriA) fails to accumulate in worms disrupted in these three mutant scenarios. Further, GFP::Brc-1(TriA) has a larger number of viable embryos than Brc1 (TriA), but not as much as in mitotic cells suggesting that accumulation in the nucleus - regardless of meiotic foci formation - can confer better DNA repair and that meiotic cells are more dependent upon Brc-1 E3 ligase function for recruitment to DNA damage. GFP::Brc-1 can partially rescue a Brd-1 null, but not the GFP::Brc-1 (TriA). Importantly, the ability to rescue loss of BRD-1 is likely dependent upon the oligomerization properties of GFP, as mScarlett-brc-1 accumulates in the nucleus but doesn’t rescue and because rescue depends upon residues proposed to be necessary for GFP oligomerization. Thus oligomerization in the nucleus is required for partial bypass of the requirement of BRD-1 in meiotic DSB repair.

This is an excellent paper that provides strong genetic evidence bolstered by some biochemistry for a structural role of Brc-1 to promote DNA damage signaling, repair, and meiotic recombination. The results are robust with one exception noted below and the authors are careful to not overstate their findings. This work provides compelling in vivo evidence for the important of BRC-1 E3 ligase activity for DNA damage and repair and provides strong support for a structural role of BRC-1 in promoting DNA repair.

Minor:

Line 164 - 181: I had trouble discerning whether the differences in pixel intensity or average number of RAD-51 foci were different from each other - particularly for whether the I123A mutant was reduced compared to zim-1 alone in 2B. E.g. I found Table S3 helpful for this. Perhaps add relevant p values for the statements in the text like “In zones 2 and 3, brc-1(I23A); zim-1 had slightly reduced numbers of RAD-51 foci compared to zim-1 (p = 0.02 and 0.0007, respectively), but significantly …

Line 180: I am not convinced the authors can argue that the structural integrity of the complex is critical with this data. The presence yes, but perhaps the structural integrity is also compromised?

Line 279: Can you provide a more detailed explanation for how the coefficient of variation (Standard deviation/mean) provides a measure of the extent of foci above the nucleoplasmic GFP signal. CV gives the variability within the sample, so could give an idea of how much of the population is above the mean?

Major:

Figure 7D - Can you provide the quantification of the mono- and polyubiquitination signal and how many times the experiment was repeated?

**Have all data underlying the figures and results presented in the manuscript been provided?**

Reviewer #1: Yes

Reviewer #2: Yes

Reviewer #3: Yes

PLOS authors have the option to publish the peer review history of their article (what does this mean?). If published, this will include your full peer review and any attached files.

Reviewer #1: No

Reviewer #2: No

Reviewer #3: No

---

## [Decision Letter · Decision Letter 1]

19 Jan 2023

Dear Dr Engebrecht,

We are pleased to inform you that your manuscript entitled "Differential requirement for BRCA1-BARD1 E3 ubiquitin ligase activity in DNA damage repair and meiosis in the Caenorhabditis elegans germ line" has been editorially accepted for publication in PLOS Genetics. Congratulations!

Yours sincerely,

Mónica P. Colaiácovo

Academic Editor

PLOS Genetics

Gregory P. Copenhaver

Editor-in-Chief

PLOS Genetics

Comments from the reviewers (if applicable):

Reviewer's Responses to Questions

**Comments to the Authors:**

Reviewer #1: The authors have addressed most of my comments, both experimentally and in the writing. I have no further issues with this manuscript and I recommend publication in PLoS Genetics. Congratulations to the authors for their excellent work.

Reviewer #2: The reviewers addressed the comments adequately.

Reviewer #3: The authors have responded well to reviewer's comments. This manuscript will be of interest to the DNA repair and reproduction communities.

**Have all data underlying the figures and results presented in the manuscript been provided?**

Reviewer #1: Yes

Reviewer #2: Yes

Reviewer #3: Yes

PLOS authors have the option to publish the peer review history of their article (what does this mean?). If published, this will include your full peer review and any attached files.

Reviewer #1: No

Reviewer #2: No

Reviewer #3: No

**Data Deposition**

http://datadryad.org/submit?journalID=pgenetics&manu=PGENETICS-D-22-01119R1

**Press Queries**

---

## [Editor Report · Acceptance letter]

25 Jan 2023

PGENETICS-D-22-01119R1 

Differential requirement for BRCA1-BARD1 E3 ubiquitin ligase activity in DNA damage repair and meiosis in the Caenorhabditis elegans germ line 

Dear Dr Engebrecht, 

We are pleased to inform you that your manuscript entitled "Differential requirement for BRCA1-BARD1 E3 ubiquitin ligase activity in DNA damage repair and meiosis in the Caenorhabditis elegans germ line" has been formally accepted for publication in PLOS Genetics! Your manuscript is now with our production department and you will be notified of the publication date in due course.

With kind regards,

Timea Kemeri-Szekernyes

PLOS Genetics

On behalf of:
